# Structure-based discovery of dual pathway inhibitors for SARS-CoV-2 entry

Haofeng Wang [1,2,12], Qi Yang [3,12], Xiaoce Liu[1,2,12], Zili Xu[4,5,12], Maolin Shao[1,2,12], Dongxu Li[1,2], Yinkai Duan [1,2], Jielin Tang[3], Xianqiang Yu[1,5], Yumin Zhang [6], Aihua Hao[7], Yajie Wang[7], Jie Chen[1,2], Chenghao Zhu[1], Luke Guddat [8], Hongli Chen[1,5], Leike Zhang [6] ✉, Xinwen Chen[3] ✉, Biao Jiang[1,5] ✉, Lei Sun [7] ✉, Zihe Rao[1,2,3,9,10,11] & Haitao Yang [1,2] ✉

Since 2019, SARS-CoV-2 has evolved rapidly and gained resistance to multiple therapeutics targeting the virus. Development of host-directed antivirals offers broad-spectrum intervention against different variants of concern. Host proteases, TMPRSS2 and CTSL/CTSB cleave the SARS-CoV-2 spike to play a crucial role in the two alternative pathways of viral entry and are characterized as promising pharmacological targets. Here, we identify compounds that show potent inhibition of these proteases and determine their complex structures with their respective targets. Furthermore, we show that applying inhibitors simultaneously that block both entry pathways has a synergistic antiviral effect. Notably, we devise a bispecific compound, **212-148**, exhibiting the dual-inhibition ability of both TMPRSS2 and CTSL/CTSB, and demonstrate antiviral activity against various SARS-CoV-2 variants with different viral entry profiles. Our findings offer an alternative approach for the discovery of SARS-CoV-2 antivirals, as well as application for broad-spectrum treatment of viral pathogenic infections with similar entry pathways.

The SARS-CoV-2 pandemic has imposed a severe public health burden worldwide since 2019. For the prevention and treatment of COVID-19, vaccines[1], antibodies[2], and antiviral medications[3–10] have been developed and approved for emergency use. However, the continuously emerging SARS-CoV-2 variants, especially Omicron subvariants, have the ability to evade the immune system when challenged by antibody-mediated neutralization or vaccine protection[11–15]. In addition, five antiviral agents have received authorization for COVID-19 treatment:

remdesivir[3], molnupiravir[4], azvudine[5], nirmatrelvir[6], and ensitrelvir[7]. The first three target the RNA-dependent RNA polymerase (RdRp), and the last two target the main protease (M[pro]). However, in vitro and in vivo studies have indicated that mutations in the SARS-CoV-2 RdRp confer resistance to remdesivir[16,17], and M[pro] mutations cause resistance to nirmatrelvir[18], suggesting the inevitable risk of drug resistance is already upon us. Unlike viral proteins, which are susceptible to mutation during SARS-CoV-2 evolution, host proteins are much more

[1]Shanghai Institute for Advanced Immunochemical Studies and School of Life Science and Technology, ShanghaiTech University, Shanghai, China. [2]Shanghai Clinical Research and Trial Center, Shanghai, P.R. China. [3]Guangzhou Laboratory, Guangzhou, China. [4]School of Physical Science and Technology, ShanghaiTech University, Shanghai, China. [5]University of Chinese Academy of Sciences, Beijing, China. [6]CAS Key Laboratory of Special Pathogens, Wuhan Institute of Virology, Center for Biosafety Mega-Science, Chinese Academy of Sciences, Wuhan, China. [7]The Fifth People's Hospital of Shanghai, Shanghai Institute of Infectious Disease and Biosecurity, and Institutes of Biomedical Sciences, Fudan University, Shanghai, China. [8]School of Chemistry and Molecular Biosciences, the University of Queensland, Brisbane, Queensland, Australia. [9]Laboratory of Structural Biology, School of Life Sciences and School of Medicine, Tsinghua University, Beijing, China. [10]State Key Laboratory of Medicinal Chemical Biology, College of Life Sciences and College of Pharmacy, Nankai University, Tianjin, China. [11]National Laboratory of Biomacromolecules, CAS Center for Excellence in Biomacromolecules, Institute of Biophysics, Chinese Academy of Sciences, Beijing, China. [12]These authors contributed equally: Haofeng Wang, Qi Yang, Xiaoce Liu, Zili Xu, Maolin Shao. ✉e-mail: zhangleike@wh.iov.cn; chen_xinwen@gzlab.ac.cn; jiangbiao@shanghaitech.edu.cn; llsun@fudan.edu.cn; yanght@shanghaitech.edu.cn

conservative and could potentially be used as therapeutic targets for the development of broad-spectrum antiviral compounds. Therefore, a viable solution for persisting drug resistance is to develop chemical compounds targeting host proteins that are indispensable to the viral life cycle.

Despite the identification of numerous host factors associated with the SARS-CoV-2 life cycle[19–23], viral entry is the primary step for infection and is already recognized as an excellent target for drug development. The entry of SARS-CoV-2 depends on two distinctive pathways, cell surface entry and endosomal entry[24], where the transmembrane serine protease, TMPRSS2, and endosomal cysteine proteases cathepsin L/B (CTSL/CTSB) cleave the viral spike protein[25,26] (Fig. 1a). Previous studies have also shown that the use of the two alternative entry pathways varies in different SARS-CoV-2 variants and their host cells[27,28]. For wild-type SARS-CoV-2 and its earlier variant of concerns (VOCs), including Delta, TMPRSS2-dependent cell surface entry is the primary pathway[29–31], whereas the Omicron variant uses endosomal entry as the primary pathway, which increases its infectivity on more cell types in the respiratory epithelium and thus enhances its intrinsic transmissibility[29]. Moreover, SARS-CoV-2 infection is sensitive to TMPRSS2 targeting inhibitors in human Calu-3 lung adenocarcinoma cells but not human A549 epithelial lung cells[27], demonstrating the polytropic preference for viral entry pathways. Therefore, to completely prevent SARS-CoV-2 infection, the simultaneous blocking of the two independent pathways for viral entry is essential[32,33].

In this study, we establish a fluorescence resonance energy transfer (FRET) assay to measure the enzymatic activity of TMPRSS2 and CTSL/CTSB, and to carry out high-throughput screening of compounds among a library of 10,000 approved drugs, clinical-trial drug candidates, and natural products. The crystal structures of TMPRSS2 and CTSL/CTSB in complex with the potent inhibitors were determined to reveal their specific mode of interaction. Notably, antiviral assays show that the combination of two compounds, nafamostat, and K777, which simultaneously blocked TMPRSS2- and CTSL/CTSB-mediated viral entry pathways exhibited synergistic antiviral activity against SARS-CoV-2, indicating that dual-inhibition is an effective strategy for clinical development to treat COVID-19. Based on the structural and antiviral information, we designed the bispecific compound **212-148**, which shows dual-inhibition of both TMPRSS2 and CTSL/CTSB. We also determine the structure of **212-148** bound to TMPRSS2 or CTSB. Cell-based assays show that this compound could prevent infection of both the Delta and Omicron variants. Our results validate that dual-inhibitors for SARS-CoV-2 entry targeting both TMPRSS2 and CTSL/CTSB are promising candidates for antiviral drug discovery.

## Results
### High-throughput screening identified potent SARS-CoV-2 entry inhibitors
To investigate the enzymatic profile of TMPRSS2, we expressed the activated catalytic ectodomain TMPRSS2 with its auto-activation

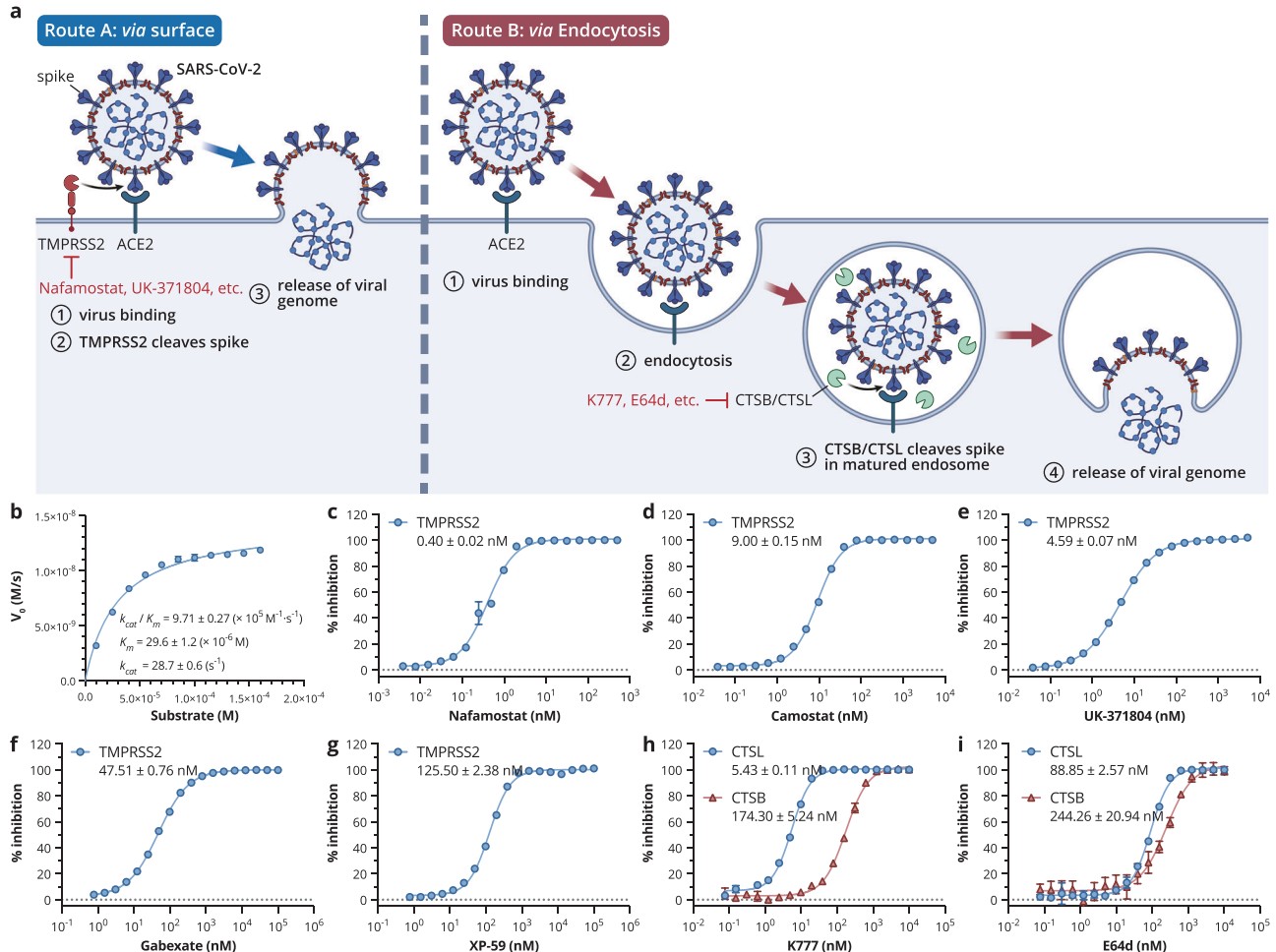

**Fig. 1 | Two independent viral entry pathways and inhibition data of compounds against two primary host proteases. a** SARS-CoV-2 can adopt two independent cell entry pathways involving host proteases. Created with BioRender.com. **b** The kinetic activity of TMPRSS2, data points are shown as mean ± SD for four biological independent replicates. **c**–**i** Dose-response curves of compounds inhibiting TMPRSS2 and CTSL/CTSB, IC50 values and data points are shown as mean ± SD for four biological independent replicates.

sequence substituted by an enteropeptidase cleavage sequence to avoid auto-cleavage of TMPRSS2[34,35]. The zymogen was purified and activated by enteropeptidase cleavage for activity assay. The enzymatic activity of TMPRSS2 was determined by a continuous kinetic assay based on FRET, using Boc-Gln-Ala-Arg-AMC as the fluorogenic substrate[36]. The catalytic efficiency ($k_{cat}/K_m$) for TMPRSS2 was measured to be 971,000 $s^{-1} \cdot M^{-1}$, demonstrating the enzyme is fully active (Fig. 1b). Subsequently, high-throughput screening of 10,000 compounds was carried out based on this enzymatic assay. Five inhibitors that showed >90% inhibition against TMPRSS2 at a concentration of 20 μM were selected for further analysis of half-maximal inhibitory concentrations ($IC_{50}$) (Fig. 1c–g). Nafamostat and camostat, two clinically available drugs used to treat pancreatitis, displayed excellent inhibitory potency with $IC_{50}$ values of 0.40 nM and 9.00 nM, respectively. Three other hits were also identified with $IC_{50}$ values ranging from ~5–125 nM. UK-371804, a drug candidate currently under preclinical evaluation for the treatment of chronic dermal ulcers[37], has an $IC_{50}$ value of 4.59 nM against TMPRSS2, a value comparable with camostat.

We used a similar procedure to screen the human CTSL/CTSB inhibitors from the same library using the fluorogenic substrate Cbz-Phe-Arg-AMC[38]. As a result, two inhibitors, K777 and E64d were identified with an $IC_{50}$ value of 5.43 and 88.85 nM for CTSL, and 174.30 and 244.26 nM for CTSB (Fig. 1h, i). Significantly, E64d has been used as a clinical drug to treat Alzheimer's disease[39].

## The structures of complete TMPRSS2 ectodomain in complex with inhibitors revealed distinctive inhibitory mechanisms

To reveal the overall architecture of TMPRSS2 and to investigate the inhibition mechanism of compounds against TMPRSS2, the activated ectodomain of TMPRSS2 was prepared for crystallization together with various inhibitors identified in the high-throughput screening. The crystal structures of TMPRSS2 in complex with nafamostat, camostat, and UK-371804, were determined to 2.6 Å, 2.4 Å and 2.6 Å, respectively.

We use the structure of TMPRSS2 in complex with camostat as the representative to describe the overall structure of TMPRSS2. There are two polypeptides in the asymmetric unit with the root-mean-square deviation (r.m.s.d) of 1.44 Å over 378 Cα atoms, indicating the two protomers are similar. The ectodomain of TMPRSS2 can be traced in the electron density map except for several small loop regions (residues 109–112, 202–205, 229–230, and 250–255). And the ectodomain of TMPRSS2 is comprised of three subdomains: an N-terminal low-density lipoprotein receptor type-A (LDLRA) domain (residues 118–148), a scavenger receptor cysteine-rich (SRCR) domain (residues 149–242), and a C-terminal trypsin-like serine peptidase (SP) domain (residues 256–492) which harbours the Ser-His-Asp catalytic triad[40,41] (Fig. 2a).

Although the same expression construct was used as in the previously reported TMPRSS2 structure[35], a major difference is observed. In the original structure, the LDLRA domain is invisible but it is clearly seen in our electron density maps. It consists mainly of random coil but there are several stabilizing loops. A canonical calcium ion stabilizes the structure by coordinating six conserved residues. Two disulfide bonds are formed around the calcium-binding region to further stabilize the overall structure of the domain (Fig. 2b). The calcium-binding motif is observed in most transmembrane serine proteases (TTSP), mediating cellular internalization of macromolecules[40]. In the absence of calcium, the LDLRA domain is unstructured, thus, the bound calcium ion is essential for structural integrity[42]. To characterize the importance of the calcium ion, we performed enzymatic and thermostability assays of TMPRSS2 in the presence of EDTA, showing that the loss of calcium ion slightly decreased the enzyme's thermal stability and catalytic activity (Supplementary Fig. 1).

The SRCR domain is another conserved protein module of TTSP. It adopts a compact fold consisting of one α-helix surrounded by a β-sheet and functions in ligand binding. A disulfide bond between Cys244-Cys365 connects the SRCR and SP chains and stabilizes the whole structure (Fig. 2a and Supplementary Fig. 2a). Although the overall structural alignment demonstrated that SRCR domains of TMPRSS1/2 exhibit the most conspicuous discrepancy, they share the same scaffold when aligned individually (Supplementary Fig. 2b, c). The SRCR domain adopts a different orientation relative to the SP domain compared to other members of the TMPRSS proteins. It is suggested that this difference may be the reason for the distinct physiological activity of each[35].

The C-terminal SP domain of TMPRSS2 forms the canonical trypsin fold with two six-stranded beta barrels. The conserved Ser441-His296-Asp345 catalytic triad is located in the central active site cleft (Fig. 2a). A structural superposition of TMPRSS2 with hepsin (TMPRSS1) showed SP domain is highly conserved, allowing us to define the S1-S4 sites involved in substrate binding (Fig. 2c and Supplementary Fig. 2d). The S1 subsite of TMPRSS2 consists of the conserved Asp435, Ser436, Gly462, Gly464, and Gly472 residues, forming a deep cavity to accommodate arginine or lysine at P1. The guanidine group of the arginine side-chain forms a diverse spread of interactions with Gly464, Ser436 and Asp435, explaining the selectivity for arginine over lysine[43]. Unlike hepsin, which prefers small and hydrophobic amino acids at P2[44], Lys342 is located at S2 subsite in TMPRSS2 and thus makes it a small and positively charged subsite, though the presence of the lysine side chain does allow some flexibility in allowing the subsite to expand. Like other members of the TMPRSS subfamily, the S3 and S4 subsites are generally wide-open with limited substrate specificity (Fig. 2c).

Both nafamostat and camostat are covalent inhibitors of TMPRSS2, adopting identical reaction mechanisms (Fig. 2d and Supplementary Fig. 3a). Initially, they bind to TMPRSS2, forming the Michaelis complex. The ester group in their common guanidinobenzoyl moiety then reacts with the catalytic serine of TMPRSS2 to form the acyl-enzyme intermediate and cleave off the "leaving group". The guanidinobenzoyl group stays covalently linked to the catalytic serine and disables the protease activity. Our structures of TMPRSS2 in complex with nafamostat and camostat both capture the post-reaction state, leaving the identical remnant moiety in the catalytic site of the SP domain. The density for the guanidinobenzoyl group is clearly defined, occupying the S1 site (Fig. 2f, g and Supplementary Fig. 4a, b). Apart from the ester bond formed with γO of the catalytic serine, the guanidyl head of the guanidinobenzoyl moiety forms a salt bridge with Asp435, and is further anchored by hydrogen bonds with Ser436 and Gly464 in the S1 pocket (Fig. 2f, g and Supplementary Fig. 5a).

UK-371804 is also a potent inhibitor of TMPRSS2, but in a non-covalent manner. Unlike nafamostat or camostat, the intact compound is observed binding to the active site (Fig. 2h). The 4-chlorine atom is directed toward the γO of the Ser441 side chain forming a hydrogen bond. UK-371804 forms salt bridges with Asp435 and hydrogen bonds with Ser436 and Gly464 residues located at the S1 pocket (Fig. 2h and Supplementary Fig. 5b). In addition, the 7-isoquinolinesulfonamides group extends to the S3 subsite although without obvious interactions, suggesting a site that could be optimized for improved potency.

## Crystal structures of CTSL/B in complex with K777 or E64d

To explore the inhibition mechanism of K777 and E64d, we determined the crystal structures of these compounds bound to human CTSL and CTSB (Fig. 3). All of these structures were solved at a high resolution with unambiguous electron density in the binding pocket of enzymes, allowing us to clearly define the interactions. Both CTSL and CTSB belong to the papain superfamily and have a typical cysteine protease fold[38,45]. The mature enzyme forms a globular structure consisting of

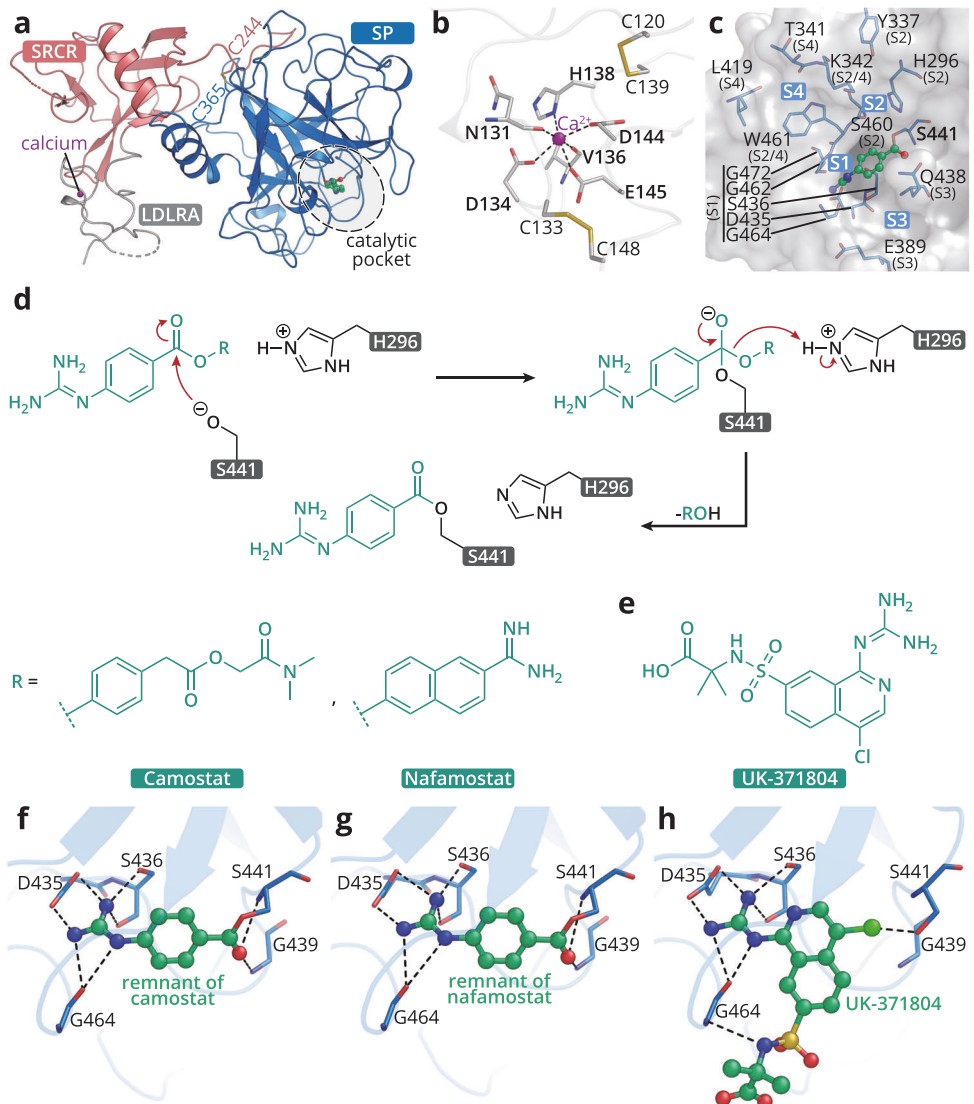

**Fig. 2 | Crystal structures of TMPRSS2 ectodomain in complex with inhibitors.** **a** Cartoon model of TMPRSS2 ectodomain in complex with camostat, different domains of TMPRSS2 are indicated and colored differently. **b** Zoom-in view of calcium-binding region located in LDLRA domain. **c** Substrate binding pocket of TMPRSS2, the surface of TMPRSS2 is colored white, residues that participates in forming S1-S4 subsites are shown as blue sticks. The inhibitor is shown as green ball- and-stick model. **d** A likely inhibition mechanism for camostat and nafamostat. **e** Skeletal structure of UK-371804. **f**–**h** Zoom-in view of the catalytic pocket of TMPRSS2 with camostat, nafamostat, or UK-371804 bound inside, compounds are shown as ball-and-stick models, residues forms hydrogen-bond or ionic bond with the compounds are shown as sticks, hydrogen-bonds and ionic bonds are shown as black dashes.

an α-helical and β-sheet domain (Fig. 3c, f). Two domains delimit an active-site cleft containing the conserved catalytic cysteine and histidine residues. In the structure of CTSL in complex with K777, this compound was found in the active site cleft. The density for K777 is connected to Cys25, evidence of a covalent bond (Supplementary Fig. 4f). The sulfonyl group is located in the oxyanion hole formed by the side chains of Gln19, Trp189, and His163 and the main chain of Cys25, whose function is to stabilize the carbonyl oxygen of the substrate at the tetrahedral intermediate stage of proteolysis (Fig. 3e and Supplementary Fig. 5d). In addition, K777 forms three hydrogen bonds with the backbone amides of Gly68 and Asp162. The phenyl sulfone, phenethyl, benzyl and N-methyl piperazine moieties of K777 insert into the S1′-S3 subsites, forming hydrophobic interactions which further stabilize the inhibitor. In the structure of CTSB in complex with K777, a comparable interaction pattern was also observed by the corresponding conserved residues except for the loss of one hydrogen bond in the oxyanion hole and the formation of one extra hydrogen

bond with the main chain of Gly198 mediated by an ordered water molecule (Fig. 3h and Supplementary Fig. 5f).

In the structure of CTSL in complex with E64d, E64d is covalently linked to the γS atom of Cys25 (Supplementary Fig. 3b). In addition, this inhibitor fills the S1-S3 subsites, forming five hydrogen bonds with the side chain of Gln19 and the backbone amide of Cys25, Asp162, and Gly68 (Fig. 3d and Supplementary Fig. 5c). The isobutyl and methylbutyl moieties of E64d contribute to the substantial hydrophobic interactions in the S2 and S3 subsites of CTSL, respectively. Notably, these two hydrophobic binding groups insert into the S2 and S3 subsites of the CTSB substrate binding pocket in a different manner. Instead of occupying in the S2 subsite of CTSL, the isobutyl moiety inserts into the S3 subsite of CTSB, leaving methylbutyl moiety in the S2 subsite, resulting in a twisted gesture of E64d (Fig. 3g and Supplementary Fig. 5e). This might be explained by the narrower S2 subsite to accommodate the smaller isobutyl group since Asp160, Met161, Asp162, and Ala214 form a continuous steric hindrance barrier in the

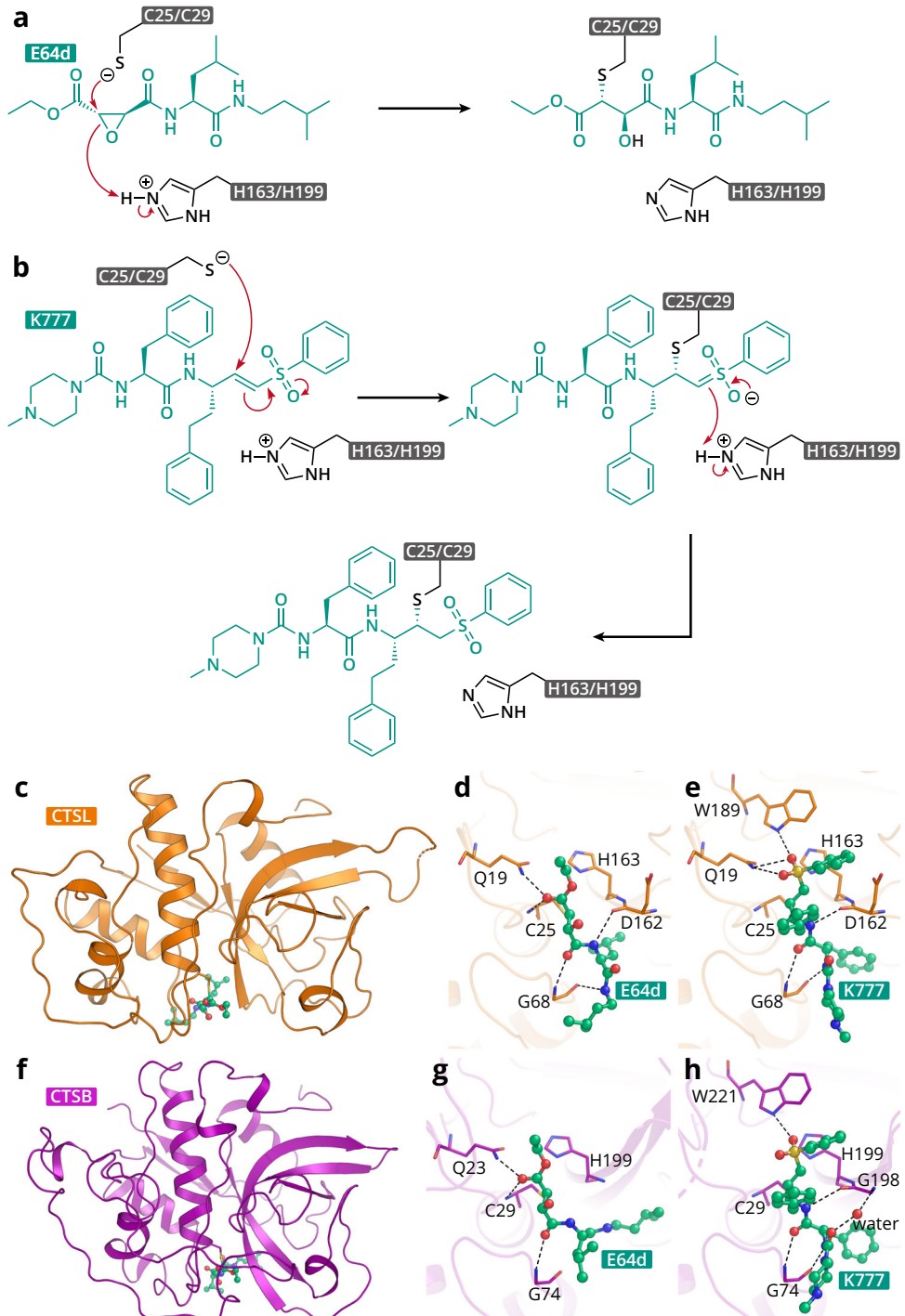

**Fig. 3 | Crystal structures of CTSL/CTSB in complex with inhibitors. a, b** Putative inhibition mechanisms of E64d and K777. **c** Overview of crystal structure of CTSL in complex with E64d. **d, e** Zoom-in view of catalytic pocket of CTSL with E64d or K777 bound, compounds are shown as ball-and-stick models, residues that form hydrogen-bond with the compounds are shown as sticks, hydrogen-bonds are shown as black dashes. **f** Overview of crystal structure of CTSB in complex with E64d. **g, h** Zoom-in view of catalytic pocket of CTSB with E64d or K777 bound, compounds are shown as ball-and-stick models, residues that form hydrogen-bond with the compounds are shown as sticks, hydrogen-bonds are shown as black dashes.

CTSL to prevent S2 subsite extension (Fig. 3g). Aside from this major difference, E64d has an almost identical mode of binding at S1 subsite in both CTSL and CTSB.

## Synergistic block of SARS-CoV-2 replication by the combination of TMPRSS2 and CTSL/CSTB inhibitors

To further validate the in vitro enzymatic inhibition results, we performed cell-based assays to evaluate the efficacy of these compounds in different SARS-CoV-2 variants and host cells. K777 and E64d, as endocytosis-mediated entrance pathway inhibitors, could effectively reduce the infectivity of the SARS-CoV-2 Omicron BA.2 variant in Calu-3 cells with an $EC_{50}$ value of 14.0 nM (Selectivity index (SI) = 7,829, SI is calculated as the ratio of $CC_{50}$ value against $EC_{50}$ value, a common measurement for comparing cytotoxicity and antiviral potency of compounds) and 239.8 nM (SI > 4,000) (Fig. 4b, d and Supplementary Fig. 6d, e). In contrast, TMPRSS2 inhibitors showed little inhibition of

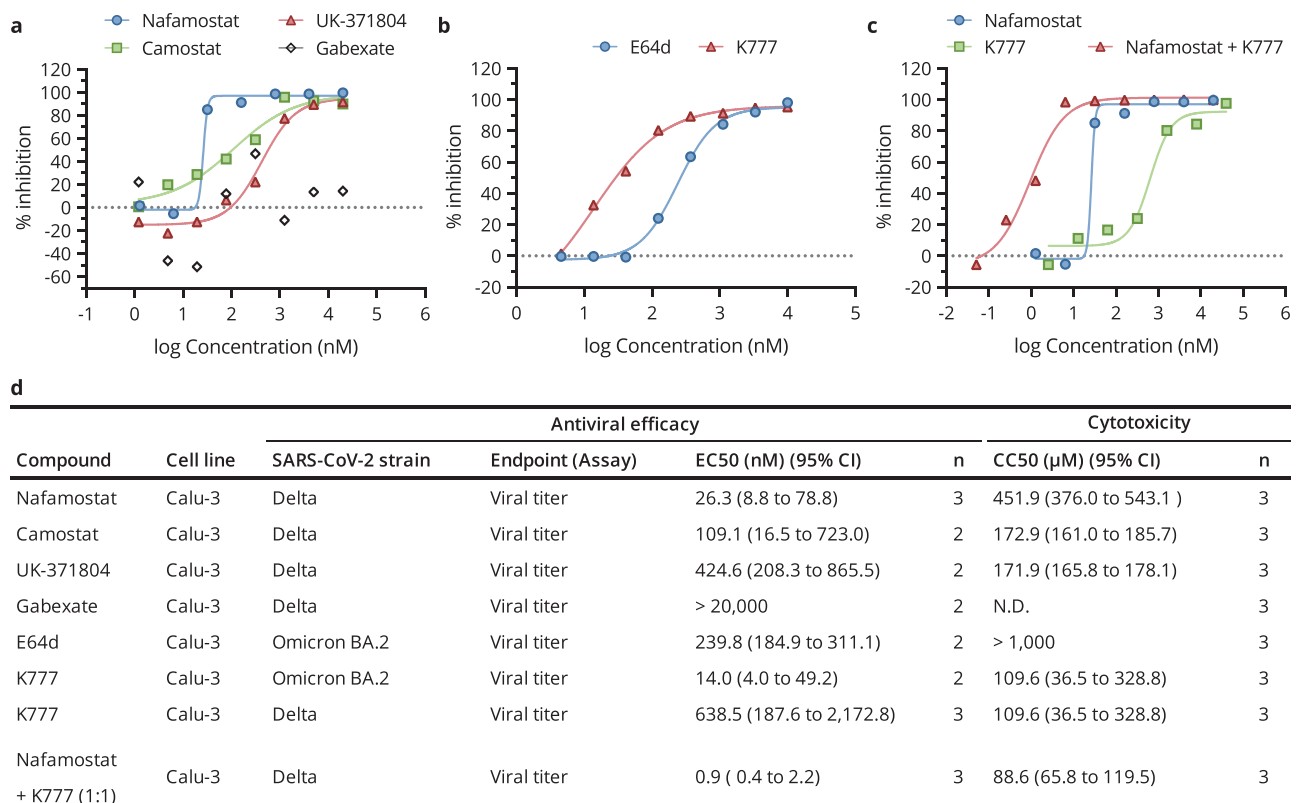

| | | Antiviral efficacy | | | | Cytotoxicity | |
|---|---|---|---|---|---|---|---|
| Compound | Cell line | SARS-CoV-2 strain | Endpoint (Assay) | EC50 (nM) (95% CI) | n | CC50 (µM) (95% CI) | n |
| Nafamostat | Calu-3 | Delta | Viral titer | 26.3 (8.8 to 78.8) | 3 | 451.9 (376.0 to 543.1 ) | 3 |
| Camostat | Calu-3 | Delta | Viral titer | 109.1 (16.5 to 723.0) | 2 | 172.9 (161.0 to 185.7) | 3 |
| UK-371804 | Calu-3 | Delta | Viral titer | 424.6 (208.3 to 865.5) | 2 | 171.9 (165.8 to 178.1) | 3 |
| Gabexate | Calu-3 | Delta | Viral titer | > 20,000 | 2 | N.D. | 3 |
| E64d | Calu-3 | Omicron BA.2 | Viral titer | 239.8 (184.9 to 311.1) | 2 | > 1,000 | 3 |
| K777 | Calu-3 | Omicron BA.2 | Viral titer | 14.0 (4.0 to 49.2) | 2 | 109.6 (36.5 to 328.8) | 3 |
| K777 | Calu-3 | Delta | Viral titer | 638.5 (187.6 to 2,172.8) | 3 | 109.6 (36.5 to 328.8) | 3 |
| Nafamostat + K777 (1:1) | Calu-3 | Delta | Viral titer | 0.9 ( 0.4 to 2.2) | 3 | 88.6 (65.8 to 119.5) | 3 |

**Fig. 4 | Antiviral activity of inhibitors targeting TMPRSS2 or CTSL/CTSB in cell-based assays. a** Calu-3 cells were pre-treated with nafamostat, camostat, gabexate or UK-371804 at different concentrations for 1 h and then infected with a clinical SARS-CoV-2 isolate Delta strain (MOI = 1). Twenty-four hours after inoculation, the supernatants were collected and virus titers were determined as TCID$_{50}$/mL. Data are shown as the geometric mean and 95% CI. The EC$_{50}$ was assessed after being cultured for three days. **b** Calu-3 cells were pre-treated with E64d or K777 at different concentrations for 1 h and then infected with a clinical SARS-CoV-2 isolate omicron strain BA.2 (MOI = 1). Twenty-four hours after inoculation, the supernatants were collected and virus titers were determined as TCID$_{50}$/mL. **c** Calu-3 cells were pre-treated with nafamostat, K777, or K777 + Nafamostat at different concentrations for 1 h and then infected with a clinical SARS-CoV-2 isolate strain Delta (MOI = 1). Twenty-four hours after inoculation, the supernatants were collected and virus titers were determined as TCID$_{50}$/mL. **d** A summary of the results, including the CC$_{50}$ values of each compound in the specific types of cells (Supplementary Fig. 6); n, the number of biological independent replicates.

Omicron infection in Calu-3 cells (Supplementary Fig. 7a–c), which is consistent with the decreased TMPRSS2 usage by Omicron variants[28]. In addition, the infectivity of the SARS-CoV-2 Delta variant, which utilizes both TMPRSS2-mediated and cathepsin-mediated membrane fusion for viral infection, could be lowered by inhibitors of either pathway. Specifically, the EC$_{50}$ value of K777 was determined to be 638.5 nM (SI = 172) and for nafamostat, camostat and UK-371804, the EC$_{50}$ values are 26.3 nM (SI = 17,183), 109.1 nM (SI = 1,585) and 424.6 nM (SI = 405), respectively(Fig. 4a, c, d and Supplementary Fig. 6a–c). The inhibitory efficacy for gabexate was negligible (Fig. 4a). In conclusion, most of the compounds inhibited their respective pathways strongly and specifically, with nafamostat and K777 being the most potent inhibitors.

To test the synergistic effect of combining both inhibitors, we infected Calu-3 cells with SARS-CoV-2 Delta variant after preincubation by combinations of nafamostat and K777 mixed at a series of molar ratios (Supplementary Fig. 7d, e). We observed that K777 and nafamostat exhibited the strongest antiviral effect at the molar ratio of 1:1. As expected, the combined use of nafamostat and K777 led to an improved inhibitory efficiency with an EC$_{50}$ value of 0.9 nM (SI = 98,444), more than 10-fold better than individual administration of either nafamostat or K777 (Fig. 4c, d and Supplementary Fig. 6f). Our data indicated a synergistic block of infection when combining nafamostat and K777, suggesting a more efficient strategy to inhibit multiple viral entrance routes.

**Bispecific compound 212-148 showed desired dual-inhibition ability in the enzymatic and antiviral assays**
In addition to the combined use of different molecules, the fusion of pharmacophoric groups into a single multi-target directed molecule is an efficient approach during the drug discovery of multifactorial diseases, which may improve PK/PD properties, reduce adverse effects and drug-drug interactions[46–49]. In this work, we intended to develop a multi-target directed drug that targets both TMPRSS2 and CTSL/CTSB simultaneously. In light of the crystal structures of K777 and nafamostat complexed with their respective proteases, we devised a bispecific compound, named **212-148**. An ethyl linker was utilized to construct this bispecific compound by connecting the pharmacophoric groups of K777 and nafamostat (Fig. 5a). We tested the enzymatic inhibition of **212-148** against TMPRSS2 and CTSL/CTSB. As expected, **212-148** not only showed potent inhibition for CTSL/CTSB with respective IC$_{50}$ values of 2.13 and 64.07 nM but also suppressed TMPRSS2 enzymatic activity with an IC$_{50}$ value of 1.38 µM (Fig. 5b, c). Further, **212-148** demonstrated effective inhibition of SARS-CoV-2 Delta variant spike protein cleavage in vitro, albeit with a decreased anti-TMPRSS2 inhibitory potency (Supplementary Fig. 8). Therefore, the designed compound **212-148** conferred the dual-inhibition effects on the biochemical assays, indicating that our drug discovery strategy of targeting two entry pathways is valid.

Further structural studies showed that **212-148** covalently linked to the TMPRSS2 catalytic site with guanidinobenzoyl moiety at the post-

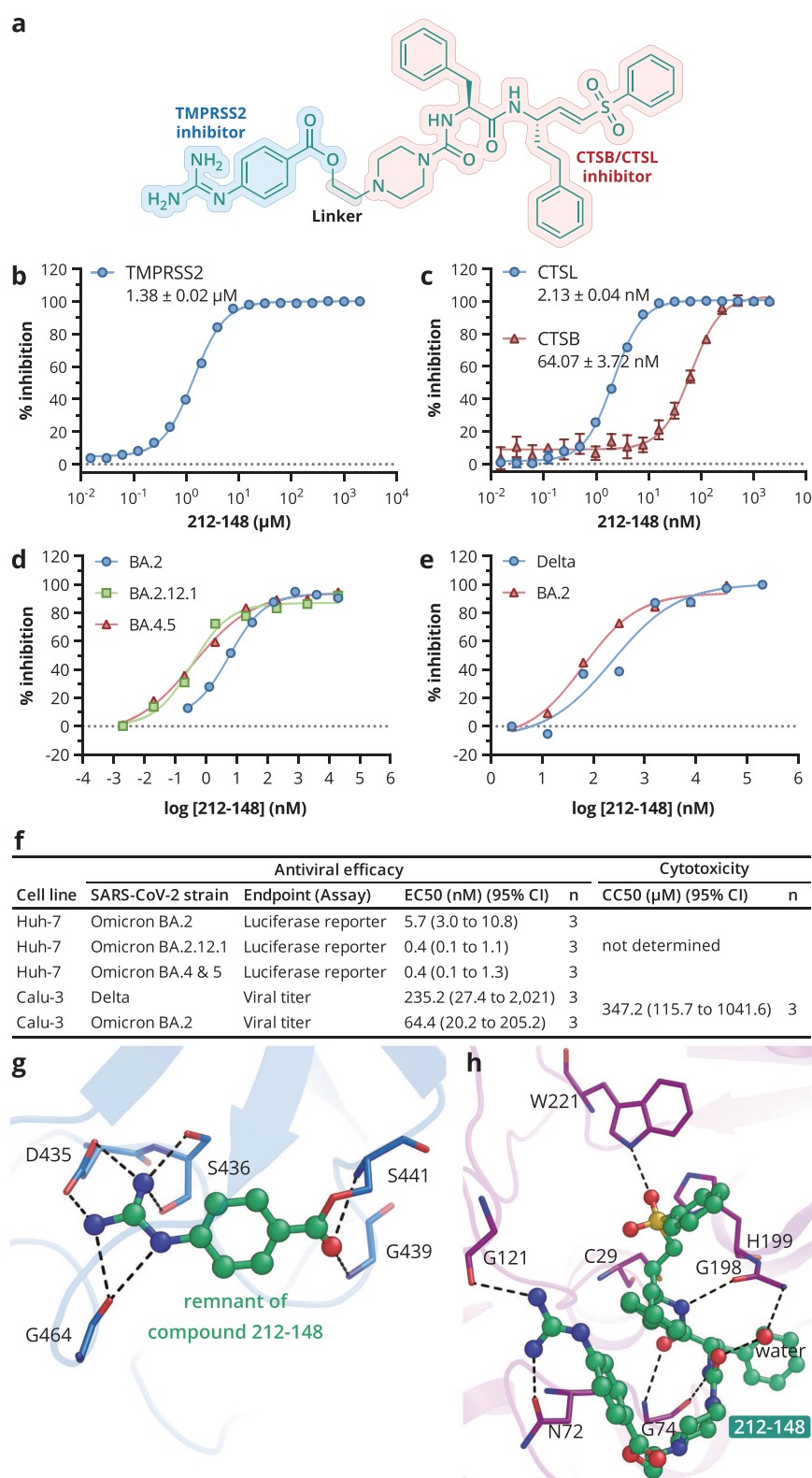

**Fig. 5 | 212-148 is a bispecific compound that blocks both viral entry pathways.**
**a** Structure of **212-148**. **b**, **c** Enzyme inhibition data for **212-148**, IC$_{50}$ values and data points are shown as mean ± SD for four biological independent replicates. (**d**–**f**) Antiviral efficacy of **212-148**. **d** Huh-7 cells pre-treated with **212-148** at different concentrations for 1 h and then infected with Omicron BA.2, BA.2.12.1 or BA.4 & BA.5 SARS-CoV-2-S pseudotyped virus (650 TCID$_{50}$/well). Chemiluminescence signals were detected twenty-four hours after the incubation of cells. **e** Calu-3 cells pre-treated with **212-148** at different concentrations for 1 h and then infected with clinical SARS-CoV-2 isolates delta strain or omicron strain BA.2 (MOI = 1). Twenty-four hours after inoculation, the supernatants were collected and virus titers were determined as TCID$_{50}$/mL. **f** A summary of the results for the antiviral experiments, including the CC$_{50}$ values of **212-148** in the specific types of cells (Supplementary Fig. 6); *n*, the number of biological independent replicates. **g** The catalytic pocket of TMPRSS2 in complex with **212-148**. **h** The catalytic pocket of CTSB in complex with **212-148**.

**f**

| | Antiviral efficacy | | | | Cytotoxicity | |
|---|---|---|---|---|---|---|
| Cell line | SARS-CoV-2 strain | Endpoint (Assay) | EC50 (nM) (95% CI) | n | CC50 (μM) (95% CI) | n |
| Huh-7 | Omicron BA.2 | Luciferase reporter | 5.7 (3.0 to 10.8) | 3 | | |
| Huh-7 | Omicron BA.2.12.1 | Luciferase reporter | 0.4 (0.1 to 1.1) | 3 | not determined | |
| Huh-7 | Omicron BA.4 & 5 | Luciferase reporter | 0.4 (0.1 to 1.3) | 3 | | |
| Calu-3 | Delta | Viral titer | 235.2 (27.4 to 2,021) | 3 | 347.2 (115.7 to 1041.6) | 3 |
| Calu-3 | Omicron BA.2 | Viral titer | 64.4 (20.2 to 205.2) | 3 | | |

reaction state, the same as nafamostat and camostat (Fig. 5g and Supplementary Fig. 5a). Unexpectedly, in the structure of **212-148** in complex with CTSB, the TMPRSS2 targeting moiety of **212-148** forms three hydrogen bonds with Gly121 and the conserved Asn72 residue located at the extended S3 subsite. This further stabilizes the ligand and explains the enhanced inhibitory effect of **212-148** on the CTSL/CTSB enzymatic activity compared with K777 (Fig. 5h and Supplementary Fig. 5g).

We then investigated the antiviral efficacy of **212-148** against SARS-CoV-2 VOCs, Omicron and Delta. In the Huh-7 cells, **212-148** inhibited the entry of pseudotyped SARS-CoV-2 Omicron variants BA.2.12.1 and BA.4 & BA.5 with a similar $EC_{50}$ value of 0.4 nM (Fig. 5d–f and Supplementary Fig. 6g), indicating the moiety of **212-148** against CTSL/CTSB continued to access the drug targets in the cell-based assays. In Calu-3 cells, **212-148** inhibits the SARS-CoV-2 Omicron BA.2 and Delta variants infection with $EC_{50}$ values of 64.4 (SI = 5,391) and 235.2 nM (SI = 1,476), respectively. Furthermore, the $TCID_{50}$ assay further indicated that viral replication could be completely blocked by **212-148**, a feat that none of the single-targeted inhibitors discussed above could achieve (Fig. 5e). In summary, our findings demonstrate a proof-of-principle validation of the possibility of simultaneously blocking two major viral entry routes with a single bispecific molecule **212-148**, offering an alternative drug discovery strategy for COVID-19 treatment and emerging SARS-CoV-2 variants.

## Discussion

The continual emergence of SARS-CoV-2 variants has posed huge challenges to treatment and protection options[50–52]. As more variants are expected to appear, it is inevitable that resistance will occur. The development of host cell-targeting antiviral therapies represents a tool to avoid drug resistance posed by new variants.

TMPRSS2 and CTSL/CTSB facilitate the entry of SARS-CoV-2 into host cells by two alternative independent pathways. As viral entry is mandatory for viral infection, these proteases have become attractive therapeutic intervention targets. Here we used a high-throughput screening approach to identify TMPRSS2 and CTSL/CTSB inhibitors from a drug library. The repurposing of known TMPRSS2 and CTSL/CTSB inhibitors can become an effective and safe treatment option for COVID-19. Nafamostat and K777, targeting TMPRSS2 and CTSL/CTSB respectively, are clinical-stage inhibitors for blocking SARS-CoV-2 infection[53,54]. Strikingly, nafamostat showed a 10-fold increase in effectiveness when combined with K777, proving that dual inhibition of these two pathways simultaneously is a more effective way to block viral infection. Moreover, comparable synergistic inhibitory outcomes were observed in the other studies[32,33] by blocking the proteases involving the viral entry pathways across different types of cells and SARS-CoV-2 variants, which further supports dual inhibition as a potential broad spectrum strategy against SARS-CoV-2 or similar infections.

The crystal structures of TMPRSS2 in complex with nafamostat or camostat in our study, and a previously reported structure of TMPRSS2 in complex with nafamostat, demonstrate a high degree of similarity, with the r.m.s.d values ranging from 0.296 to 0.445 Å for all Cα atoms of their SP domains (residues 256–492). The values are low because only the same remnant moiety of guanidinobenzoyl group occupies the active site of TMPRSS2 after the reaction, although the complete structures of nafamostat and camostat are different (Fig. 2d). These TMPRSS2-inhibitor complex structures, as well as the structures of CTSL/CSTB in complex with inhibitors, delineate the active site pocket of the enzymes and revealed the inhibition mechanisms, providing a guide for improving inhibition. Based on the structural analysis and synergistic inhibition assays, we devised a bispecific compound, **212-148**, by linking the covalent moieties of K777 and nafamostat together. Ziprasidone, an FDA-approved drug created by combining dopamine and a 5-HT$_2$ antagonist, was able to simultaneously bind two targets, the D$_2$ and 5-HT$_{2A}$ receptors, demonstrating the effectiveness of this strategy for the development of multi-target

directed molecules[55–57]. Our integrated structural, biochemical, and antiviral data have verified **212-148**'s dual-inhibition functionality in the two alternative viral entry pathways. Although its anti-TMPRSS2 inhibitory activity decreased in comparison to nafamostat, **212-148** showed nanomolar potency against Omicron and Delta variants while maintaining good cell viability. Certainly, further modification of this bispecific compound may be required to achieve optimal potency and pharmacokinetics for a potential clinical study. In conclusion, this bispecific compound has validated the concept of blocking two major viral entry routes with a single molecule, posing an innovative and practical structure-based drug discovery strategy for anti-SARS-CoV-2 therapeutics.

Previous studies have suggested that TMPRSS2 and CTSL/CTSB inhibitors effectively prevent the infection of other coronaviruses, including SARS-CoV, MERS-CoV, and human coronavirus (HCoV)-229E[58–60]. Therefore, dual-inhibition of TMPRSS2 and CTSL/CTSB offers a broad-spectrum anti-coronavirus strategy that is effective not only for the current SARS-CoV-2 pandemic but also for other potential coronavirus pandemics.

Other than fighting against coronavirus, the identified TMPRSS2 inhibitors can also be used to treat other diseases related to TMPRSS2. Entry of influenza virus is dependent on the TMPRSS2-mediated cleavage of the viral surface glycoprotein precursor HA0, making TMPRSS2 a key antiviral target. Therefore, drug development targeting TMPRSS2 also provides us with alternative solutions to treat influenza. In addition, coinfection with influenza virus (Flurona) would enhance SARS-CoV-2 infectivity[61,62], which challenges the current antiviral treatment targeting the viral protease specific to one virus. Host protease TMPRSS2-based treatment could be a feasible strategy to cure Flurona. Our crystal structure of the complete TMPRSS2 ectodomain presented here provided a solid basis for further drug design and development.

Although we have focused on dual inhibition of TMPRSS2 and CTSL/CTSB involved in host cell entry, our strategy is also applicable to the combination of inhibitors targeting different host targets involved in host entry and host antiviral response. Our findings also apply to combining different inhibitors targeting host and targeting virus, such as RdRp, M$^{pro}$ and PL$^{pro}$. As evidenced in the drug development for HIV or HCV, using an inhibitor cocktail is a highly effective strategy to treat infectious diseases[63,64]. The lead compounds we have identified here can serve as promising leads for use in combination with other antivirals against SARS-CoV-2.

## Methods
### DNA manipulation
The coding gene for human TMPRSS2 was synthesized and codon-optimized for expression in the *Spodoptera frugiperda* (sf9) cells by the GENEWIZ. The PCR fragment containing ectodomain of TMPRSS2 (residues 109–492) was amplified and inserted into the pFastBac1 vector using restriction sites *BamH* I and *Xho* I, possessing an N-terminal GP64 (baculovirus envelope glycoprotein) signal peptide and C-terminal 10 × His tag. The sequence responsible for TMPRSS2 autoactivation ($_{250}$SSRQSR$_{255}$) was substituted with the enteropeptidase cleavage site (DDDDK). The final plasmid was transformed into *E. Coli* (DH10 Bac strain) competent cells to generate the recombinant bacmid DNA.

The genes coding human CTSL (residues 18–333) and CTSB (residues 18–333) were synthesized from GenScript and the sequence was codon-optimized for expression in *E. coli*. The DNA segments of CTSL and CTSB were amplified and subcloned into the pET28a vector using restriction sites *Nco* I/*BamH* I and *Nde* I/*BamH* I, respectively.

### Protein expression, purification, and activation of TMPRSS2
The resulting bacmid was transfected into sf9 cells at a cell density of $2.0 \times 10^6$ cells/ml using Cellfectin II Reagent (Invitrogen) according to

the manufacturer's instructions. After 72 h post-infection, P0 viral stock was collected and continually amplified for the production of higher-titer P1 to P3 viral stock. High Five cells were selected for TMPRSS2 overexpression. Four liters of High Five cells cultured in the Sf-900™ II SFM (Thermo Fisher Scientific) were infected with the P3 virus at a cell density of $2.5 \times 10^6$ cells/ml. TMPRSS2 was expressed and secreted outside cells for 4 days after baculovirus infection at 27 °C under constant shaking.

Cell culture supernatant was harvested by centrifugation (4 °C, $20000 \times g$, 15 min) to remove the cell pellet. The supernatant was concentrated to 300 mL and diluted with TBS (25 mM Tris, pH 8.0, 150 mM NaCl), followed by Ni-NTA affinity chromatography purification. The protein bound to the Ni-NTA column was washed using TBS supplemented with 30 mM imidazole before elution using TBS supplemented with 250 mM imidazole. The eluted sample was concentrated to 2 mg/ml and exchanged to Reaction buffer (25 mM Tris, pH 8.0, 150 mM NaCl, 2 mM CaCl₂), which is suitable for enteropeptidase (NEB) digestion to activate the TMPRSS2 zymogen. After incubation with enteropeptidase (15 U/mg TMPRSS2) and PNGase F (NEB; 150 U/mg TMPRSS2) overnight at 20 °C, the activated sample was concentrated to load onto the Superdex 75 Increase 10/300 GL column (GE Healthcare, USA) in gel filtration buffer (25 mM Tris, pH 8.0, 75 mM NaCl). The pooled fractions containing the target protein were concentrated and stored at −80 °C for further use.

### Production, denaturation and refolding of CTSL/CTSB inclusion bodies

Both CTSL/CTSB were overpressed as inclusion bodies in the *E. coli*, and subjected to a similar denaturation and refolding condition to produce the viable enzymes. Firstly, the recombinant plasmids containing CTSL/CTSB were transformed into *E. coli* BL21(DE3) cells and then bacteria were cultured in the Luria broth (LB) at 37 °C until the $OD_{600}$ approached 0.6-0.8. Overexpression of proteins was induced by the addition of 0.4 mM IPTG (isopropyl thio-β-D-galactoside). Then the harvested cells were resuspended in lysis buffer (50 mM Tris, pH 8.0, 5 mM EDTA, 5% sucrose, 2 mM Protease Inhibitor Cocktail) and disrupted by the high-pressure homogenizer at 4 °C. After centrifugation, the inclusion bodies of CTSL/CTSB were collected and sequentially washed with buffer A (50 mM Tris, pH 8.0, 5 mM EDTA, 0.1% Triton X−100) and buffer B (50 mM Tris, pH 8.0, 2 mM EDTA, 2 M urea). Subsequently, the final inclusion bodies were denatured in buffer C (50 mM Tris, pH 8.0, 5 mM EDTA, 8 M urea, 150 mM NaCl, 5 mM DTT(dithiothreitol)) at a concentration of 2.5 mg/mL and shaken vigorously for 1 h.

The refolding of CTSL/CTSB was performed using a sample pump with a flow rate of 0.1 mL/min to drip solubilized inclusion bodies into 3 L refolding buffer (50 mM Tris, pH 8.5, 0.5 M L-Arginine, 0.01% v/v BRIJ 35, 10 mM NaCl, 100 μg/L Catalase (200000 unit/g), 10 mM GSH, 1 mM GSSG) at 4 °C, stirred simultaneously overnight. Then the solution containing folded enzymes was filtered, concentrated and dialyzed against dialysis buffer (50 mM Tris, pH 8.0, 500 mM NaCl) at 4 °C.

### Purification and activation of human CTSL/CTSB

After refolding, the CTSL/CTSB adopted a different strategy for sequent purification and activation. The folded CTSL solution was brought to a final concentration of 1.2 M $(NH_4)_2SO_4$ before purification through hydrophobic interaction chromatography (Phenyl Beads 6FF column (High Sub), Cytiva, USA) using linear gradient elution. Fractions containing CTSL were pooled and incubated in the activation buffer A (100 ng/mL sodium dextran sulfate, 100 mM sodium acetate, pH 5.0, 1 mM DTT) for 1 h at 37 °C to auto-activate the zymogen. For CTSB, the protein solution was loaded on HiTrap Q column equilibrated with Buffer A (20 mM Tris/HCl, pH 8.0) and eluted with a linear gradient to Buffer B (20 mM Tris/HCl, 1 M NaCl, pH 8.0). Fractions

containing CTSB were collected and adjusted to pH 3.5 by adding 1 M formic acid and pepsin at the molar ratio of 1: 100 relative to CTSB for activation. Finally, the activated CTSL/CTSB was concentrated and loaded on a Superdex 75 Increase 10/300 GL column (GE Healthcare, USA) in gel filtration buffer (20 mM Bis-Tris methane, pH 7.0, 100 mM NaCl, 5 mM DTT, 5% glycerol). The pooled fractions containing the target protein were concentrated and stored at −80 °C for further use.

### Crystallization, data collection and structure determination

The crystallization was performed with purified TMPRSS2 (8 mg/mL) incubated with camostat, nafamostat, UK-371804 or **212−148** at a molar ration of 3:1 for 2 h on ice. Crystals for TMPRSS2 in complex with various inhibitors were grown by hanging drop vapor diffusion at 20 °C, with the same crystallization reservoir solution containing 0.1 M acetic acid/sodium acetate, pH 5.0, and 16% w/v PEG 8000.

The purified and activated CTSL at a concentration of 8 mg/mL was incubated with E64d or K777 at a molar ration of 5:1 for 2 h at room temperature. Crystals were obtained by sitting drop vapor diffusion at 16 °C, with the reservoir solution consisting of 100 mM citric acid, pH 3.5 and 3 M sodium chloride for CTSL-E64d complex; the reservoir solution consisting of 9% (v/v) 2-propanol, 90 mM sodium cacodylate/ hydrochloric acid pH 6.5, 180 mM zinc acetate, and 0.5%w/v n-dodecyl-N,N-dimethylamine-N-oxide (LDAO, DDAO) for the CTSL-K777 complex.

The purified and activated CTSB at a concentration of 5.5 mg/mL was incubated with E64d, K777 or **212-148** at a molar ration of 10:1 overnight at 4 °C. Crystals were obtained by sitting drop vapor diffusion at 16 °C, with the reservoir solution consisting of 0.2 M ammonium acetate, 0.1 M sodium acetate trihydrate (pH 4.6) and 30% w/v polyethylene glycol 4,000 for CTSB-E64d complex; the reservoir solution consisting of 17.1 % v/v polyethylene glycol 600, 50 mM MES (pH 5.6) and 8.6 % w/v polyethylene glycol 4,000 for CTSB-K777 complex; the reservoir solution consisting of 0.2 M magnesium chloride hexahydrate, 0.1 M sodium acetate (pH 5.2) and 28 % w/v PEG 3350 for CTSB-**212-148** complex. All the crystals above were cryoprotected by reservoir solutions supplemented with 20% glycerol and flash frozen in liquid nitrogen for further data collection.

The X-ray diffraction data were collected at beamlines BL18U1, BL19U1, BL02U1 and BL10U2 at the Shanghai Synchrotron Radiation Facility (SSRF), China and beamline I04 of Diamond Light Source, UK. Data were indexed, integrated and scaled with XDS[65]. For structures of TMPRSS2 in complex with inhibitors, the phases were determined by the molecular replacement using a coordinate file of TMPRSS2 in complex with nafamostat (PDB ID: 7MEQ) as a search model in Phenix[66]. Similarly, the structures of CTSL/CSTB in complex with inhibitors were determined by molecular replacement using models (PDB ID: 6EZX) and (PDB ID: 5MAJ) as templates, respectively. Subsequently, all the models were subjected to iterative cycles of refinement with Phenix[66]. The inhibitors were built manually according to the omit map with *Coot*[67]. Data collection and structure refinement statistics are summarized in Tables S1–3.

### Enzymatic activity assay

The enzymatic activity of TMPRSS2 was measured by a continuous kinetic assay based on FRET, using the fluorogenic substrate Boc-Gln-Ala-Arg-AMC (GL Biochem). The excitation and emission wavelengths of the fluorogenic substrate were 340 nm and 460 nm, respectively. Fluorescence intensity was monitored with an EnVision multimode plate reader (Perkin Elmer). The reaction assay was performed at a final volume of 50 μL in the buffer containing 20 mM Tris pH 7.4, 150 mM NaCl, 0.1 mg/ml BSA. The concentration of TMPRSS2 was set to 0.5 nM. The reaction was initiated by adding TMPRSS2 into a solution containing different concentrations of substrate (10–160 μM) and their fluorescence values were recorded immediately. Initial velocities were calculated by fitting the linear portion of the curves to a straight line.

The data were analyzed by non-linear regression analysis in GraphPad Prism version 9.4 (Dotmatics) to generate the kinetic parameters $K_m$ and $k_{cat}$.

## High-throughput drug screening

Potential inhibitors against human TMPRSS2 were screened by an enzymatic inhibition assay in vitro. High-throughput drug screening was performed on about 10000 compounds from five drug libraries, the Approved Drug Library (Target Mol), Clinic Compound Library (Target Mol), FDA-approved Drug Library (Selleck), Natural Product Library (Selleck), and Anti-virus Drug Library (Shanghai Institute for Advanced Immunochemical Studies). The screening assays were performed in 384-well black microplates (PerkinElmer) at a total volume of 50 µL. Bravo Automated Liquid Handling Platform (Agilent) was used to rapidly add the compounds into the enzymatic reaction mixture. The final reaction system adopted the enzymatic assay buffer supplemented with 0.1 mg/mL BSA, 0.01% v/v Triton X-100, including 0.5 nM TMPRSS2, 10 µM substrate, and 10 µM compounds. DMSO treatment was used as a negative control. The initial velocities were changed by the addition of compounds, and compared with the control to evaluate their inhibitory effect. The compounds with inhibition over 90% were defined as hits and picked up for further testing. The CTSL/CTSB adopted a similar high-throughput drug screening approach with their specific substrate (Cbz-Phe-Arg-AMC, GL Biochem) at a final concentration of 10 µM. The reaction systems also used their enzymatic conditions supplemented with 0.1 mg/mL BSA, 0.01% v/v Triton X-100. All experimental statistic was analyzed using GraphPad Prism 9.4.

## IC$_{50}$ measurement

TMPRSS2 was diluted in IC$_{50}$ assay buffer (20 mM Tris-HCl pH 7.4, 150 mM NaCl, 0.1 mg/mL BSA, 0.01% v/v Triton X-100) and the IC$_{50}$ values of TMPRSS2 inhibitors were measured using 0.5 nM TMPRSS2 and 10 µM substrate in the presence of different concentrations of compounds. Fluorescence intensity was monitored with an EnVision multimode plate reader (Perkin Elmer) and the initial velocities were obtained by fitting the linear portion of the curves to a straight line. IC$_{50}$ was derived by fitting a nonlinear regression curve in GraphPad Prism version 9.4 (Dotmatics). For CTSL/CTSB, the general process is similar except the buffer was changed to 0.4 M acetic acid/sodium acetate pH 5.5, 4 mM EDTA, 5 mM DTT, 0.1 mg/mL BSA, 0.01% v/v Triton X-100.

## Thermal stability assay

Using the Prometheus NT.48 instrument from NanoTemper Technologies, we conducted the real-time simultaneous monitoring of intrinsic tryptophan fluorescence (ITF) at 330 nm and 350 nm, and the excitation wavelength is 280 nm[68]. The measurements were performed using 10 µL of a suspension of TMPRSS2 (0.17 mg/mL in 20 mM Tris-HCl pH 7.4, 150 mM NaCl, 2 mM CaCl$_2$, with or without 10 mM EDTA) in a capillary placed in the sample holder. The temperature was increased from 20 to 90 °C at a ramp rate of 1 °C/min, with one fluorescence measurement taken at every 0.025 °C increments. The ratio of the recorded emission intensities (Em350nm/Em330nm) was plotted as a function of temperature, and the fluorescence intensity ratio and its first derivative was calculated using the manufacturer's software (PR.ThermControl, version 2.3.1). This ratio represents the change in TRP fluorescence intensity and the shift in emission maximum to higher wavelengths ("red-shift") or lower wavelengths ("blue-shift"). For each condition, we conducted three independent measurements.

## SARS-CoV-2 spike protein cleavage inhibition assay

Spike protein of the SARS-CoV-2 Delta variant was cloned, expressed and purified as reported previously[69,70]. In brief, the gene encoding spike with the substation of proline at residues 986 and 987 was synthesized and inserted into the pcDNA3.1 vector for expression. After expression for 3 days, the supernatant was collected, and soluble spike protein was purified by Ni-NTA affinity chromatography. Spike protein was further purified via gel filtration chromatography with a Superose 6 10/300 column (GE Healthcare, USA) in a PBS buffer (pH 7.4). Spike protein lacking the S1/S2 site was obtained through the replacement of the $_{682}$RRAR$_{685}$ with GSAS. Both spike proteins with or without the S1/S2 site were concentrated at 1.5 mg/mL and incubated with 1.5 µM TMPRSS2, respectively, in the assay buffer (20 mM Tris pH 8.0, 150 mM NaCl). For cleavage inhibition assay, TMPRSS2 was pre-incubated with 10 times molar **212-148** for 1 h. When digested at 0, 1,5,15, 30, and 60 min, the SDS-PAGE samples under each condition were immediately prepared and further visualized in the gels by Coomassie blue. Additionally, nafamostat served as a positive control, and SARS-CoV-2 M$^{pro}$ served as a negative control in these assays.

## Cell culture

Huh-7 (JCRB, 0403) cells were cultured in Dulbecco's Modified Eagle Medium (DMEM); Human Calu-3 (ATCC, HTB-55) cells were cultured in Minimum Essential Medium (MEM). All media were supplemented with 10% fetal bovine serum (FBS) and containing 100 IU/mL penicillin and 100 µg/mL streptomycin. All cells were cultured at 37 °C in a fully humidified atmosphere containing 5% CO$_2$, and have been tested negative for mycoplasma infection.

## Virus preparation and titrations

SARS-CoV-2 Delta variant (Delta-IM2175251-P3-YQ-500 µL, Delta), Omicron BA.1.1 variant (Omicron-BA.1.1-IM21Y6017-P4-YQ-250 µL, Omicron BA.1.1) and Omicron BA.2-3 variant (Omicron-BA.2-3-P2-YQ-500 µL, Omicron BA.2) were propagated in Vero E6 cells (ATCC, CRL-1586). Virus titers were determined with 10-fold serial dilutions in confluent Vero E6 cells in 96-well microtitre plates. Three days after inoculation, a cytopathic effect (CPE) was scored, and the Reed-Muench formula was used to calculate the TCID$_{50}$. SARS-CoV-2 Delta, Omicron BA.1.1 and BA.2 stocks used in the experiments had undergone three, four and two passages on Vero E6 cells and were stored at −80 °C, respectively. All of the infection experiments were performed at BSL-3 in Guangzhou Customs Inspection and Quarantine Technology Center (IQTC).

## Cell viability assay

Cell viability was evaluated using a Cell Titer-Glo Luminescent Cell Viability Assay kit (Promega) according to the manufacturer's instructions. In brief, $2 \times 10^4$ cells in 100 µl culture medium were seeded into opaque-walled 96-well plates for 24 h and 100 µl of Cell Titer-Glo reagent was added to each well. After 5-minutes shaking and 10-minutes incubation, luminescence was measured by GloMax 20/20 (TurnerBio Systems). The half-cytotoxic concentration (CC$_{50}$) was assessed in the absence of viruses after being cultured for one or two days.

## Pseudotyped SARS-CoV-2 assay

SARS-CoV-2-S pseudotyped virus Omicron BA.2 (DR-XG-C011), Omicron BA.2.12.1 (DR-XG-C015) and Omicron BA.4/BA.5 (DR-XG-C013) were purchased from Guangzhou DARUI Biotechnology Co., Ltd. The VSV-based pseudotyped SARS-CoV-2 variants were produced by transfecting 293 T cells (ATCC, CRL-3216) with spike protein expression plasmids and simultaneously infected with G*ΔG-VSV (Kerafast, Boston, MA). $2 \times 10^4$ Huh-7 cells were seeded in a 96-well plate and pretreated with different doses of the compounds for one hour. The compounds with different dilution concentrations were mixed with SARS-CoV-2 (650 TCID$_{50}$/well), and 200 µL mixtures were inoculated onto monolayer Huh-7 cells. Chemiluminescence signals were detected twenty-four hours after the incubation of cells and virus at 37 °C with 5% CO$_2$. The Britelite plus reporter gene assay system

(PerkinElmer, Waltham, MA) and PerkinElmer Ensight luminometer were used for signal collection. The inhibition of compounds and the values of $EC_{50}$ are calculated from the luciferase level of pseudotyped SARS-CoV-2. Two independent experiments were performed with triplicate infections and one representative is shown.

## Authentic SARS-CoV-2 assay

$5 \times 10^5$ Calu-3 cells were seeded in a 24-well plate and pre-treated with different doses of the compounds for one hour. The compounds with different dilution concentrations were mixed with SARS-CoV-2 (MOI = 1), and 500 µL mixtures were inoculated onto Calu-3 cells. Two hours after inoculation, the wells were extensively washed with PBS, and then inoculated with different dilution concentrations compounds for twenty-four hours. The supernatants were collected and virus titers were determined with 10-fold serial dilutions in confluent Vero E6 cells in 96-well microtitre plates. Three days after inoculation, CPE was scored using Celigo Image Cytometer, and the Reed-Muench formula was used to calculate the $TCID_{50}$. The inhibition of compounds and the values of $EC_{50}$ were calculated from SARS-CoV-2's titers. Three independent experiments were performed with triplicate infections and one representative is shown.

## Synthesis of compound 212-148

Compound **212-148** was synthesized through ten steps reaction as shown in the scheme (Supplementary Fig. 9). **212-148** was fully characterized by NMR and MS.

All chemical reagents were of analytical grade, obtained from commercial sources, and used as supplied without further purification unless indicated. NMR spectra were recorded on a Bruker-500 (500 MHz) instrument. The deuterated solvents employed were purchased from Energy Chemical. Chemical shifts were given in ppm with respect to referenced TMS peaks. Spectra were analyzed with MestReNova. High-resolution mass spectra (HRMS-ESI) were obtained on an ABsciex 4600.

**Synthesis of 212-111.** To a solution of boc-L-homophenylalanine (5.59 g, 20 mmol) in THF (100 ml), EDCI (4.6 g, 24 mmol), HOBt (3.24 g, 24 mmol) and DIPEA (12.92 g, 100 mmol) were added. This mixture was cooled to 0 °C and stirred for 10 min. Then N,O-Dimethylhydroxylamine hydrochloride (2.34 g, 24 mmol) was added. The reaction mixture was diluted by water after stirring overnight and extracted by EA. The organic layer was washed with saturated NaHCO₃ solution and dried by sodium sulfate. Solvent was removed under vacuum. The residue was purified by Flash Column Chromatography (using EA in PE from 20%–50%) to obtain the desired product **212-111**[71]. Yield = 67.8%

**Synthesis of 212-114.** To a cooled solution of **212-111** (3.2 g, 10 mmol) in dry THF, LiAlH₄ (0.45 g, 12 mmol) was added slowly within 10 min under 0 °C. The reaction mixture was stirred for 30 min under this temperature. After quenched by H₂O, diluted HCl (1 M) was added to adjust pH to 6. The mixture was extracted by EA for 3 times. The organic layer was collected and washed by saturated NaHCO₃ solution, and dried by sodium sulfate. After removing the solvent by vacuum, the residue was purified by Flash Column Chromatography (using EA in PE from 20%–50%) to obtain the desired product **212-114**[71]. Yield = 89%.

**Synthesis of 212-117.** To a suspension of NaH (60% in mineral oil, 0.427 g, 10.678 mmol) in THF (60 ml), the solution of diethyl p-[(phenylsulfonyl)methyl]phosphonate (2.861 g, 9.79 mmol) in THF (10 ml) was added dropwise under 0 °C. This reaction mixture was stirred for 30 min under this temperature. **212-114** was then added to the reaction mixture and allowed to stir for a further hour. The mixture was concentrated under vacuum. The residue was diluted by H₂O and

extracted by EA. The organic layer was collected and washed by saturated NaHCO₃ solution, brine respectively and further dried by Na₂SO₄. The residue was purified by Flash Column Chromatography (using EA in PE from 20%–50%) to obtain the desired product **212-117**[71]. Yield = 53.8%

**Synthesis of 212-117-2.** To a solution of **212-117** (2.113 g, 5.27 mmol) in DCM (5 ml), TFA (5 ml) was added. This reaction mixture was stirred for 3 h and the solvents were removed under vacuum. The crude product was purified by HPLC. Yield = 93.6%. ESI-HRMS calcd for $C_{17}H_{29}NO_2S$ [(M + H)⁺]: 302.1215 found: 302.3073. ¹H NMR (500 MHz, MeOD) δ 7.99 − 7.93 (m, 2H), 7.77 − 7.70 (m, 1H), 7.65 (dd, J = 8.5, 7.1 Hz, 2H), 7.25 (dd, J = 8.1, 6.8 Hz, 2H), 7.21 − 7.14 (m, 1H), 7.13 (dd, J = 7.0, 1.7 Hz, 2H), 7.01 (d, J = 15.2 Hz, 1H), 6.94 (dd, J = 15.2, 7.4 Hz, 1H), 4.03 (td, J = 8.2, 5.6 Hz, 1H), 2.68 − 2.53 (m, 2H), 2.16 (m, 1H), 2.11 − 1.99 (m, 1H). ¹³C NMR (126 MHz, MeOD) δ 139.61, 139.56, 139.30, 135.73, 133.97, 129.48, 128.38, 127.99, 127.70, 126.19, 50.85, 33.81, 30.83 (Supplementary Fig. 10).

**Synthesis of 212-108.** To a solution of L-Phenylalanine methyl ester (215.7 mg, 1 mmol) in the mixture of DCM (2 ml) and saturated NaHCO₃ solution (2 ml), triphosgene (97.9 mg, 0.33 mmol) was added in one portion under 0 °C. This reaction mixture was stirred for 15 min under this reaction. The organic layer was collected and washed with brine, dried (Na₂SO₄) and concentrated under vacuum. The residue was used immediately in the next step without any further purification.

The crude product in THF (3 ml) was added to a solution of 1-piperazineethanol (114.1 mg, 0.80 mmol) and DIPEA (206.0 mg, 1.59 mmol) in THF (3 ml) under 0 °C. The resulting solution was stirred under room temperature overnight. THF was removed under vacuum and the residue ws purified by Flash Column Chromatography (using MeOH in DCM from 0%–10%) to obtain the desired product **212-111**. Yield = 42%. ESI-HRMS calcd for $C_{17}H_{26}N_3O_4$ [(M + H)⁺]: 336.1923 found: 336.3982.

**Synthesis of 212-125.** To a solution of **212-108** (286.5 mg, 0.855 mmol) in THF (10 ml), LiOH • H₂O (107.7 mg, 2.57 mmol) in H₂O (3 ml) was added slowly. The reaction mixture was monitored by TLC. After the reaction, HCl in dioxane (4 M) was added to adjust pH to 2 under 0 °C. Solvent was removed under vacuum and the residue was washed by ethyl ester. Organic solution was collected and concentrated under vacuum. The crude product was purified by HPLC. Yield = 79.0% ESI-HRMS calcd for $C_{16}H_{24}N_3O_4$ [(M + H)⁺]: 322.1767 found: 322.3612. ¹H NMR (500 MHz, MeOD) δ 7.30 − 7.22 (m, 4H), 7.17 (ddd, J = 8.6, 5.9, 2.3 Hz, 1H), 4.40 (dd, J = 8.7, 4.9 Hz, 1H), 3.88 (dd, J = 6.3, 4.1 Hz, 2H), 3.70 − 3.56 (m, 5H), 3.20 (dt, J = 12.5, 6.3 Hz, 1H), 3.18 − 3.07 (m, 6H), 3.01 (dd, J = 13.7, 8.7 Hz, 1H). ¹³C NMR (126 MHz, MeOD) δ 177.59, 157.50, 138.57, 129.18, 127.97, 126.12, 66.78, 58.75, 57.25, 55.83, 51.78, 41.15, 37.81 (Supplementary Fig. 11).

**Synthesis of 212-127.** To a solution of **212-125** (274.5 mg, 0.855 mmol) in DMF (8 ml), EDCI (163.34 mg, 191.7 mmol), HOBt (115.53 mg, 0.855 mmol), and DIPEA (442 mg, 3.42 mmol) were added. This reaction mixture was stirred overnight and purified by HPLC directly to obtain the desired product **212-127**. Yield = 32.8%. ESI-HRMS calcd for $C_{33}H_{41}N_4O_5S$ [(M + H)⁺]: 605.2798 found: 605.4355. ¹H NMR (500 MHz, MeOD) δ 8.22 (d, J = 8.2 Hz, 1H), 7.88 − 7.82 (m, 2H), 7.73 − 7.65 (m, 1H), 7.61 (dd, J = 8.5, 7.1 Hz, 2H), 7.25 − 7.19 (m, 6H), 7.15 (tq, J = 6.1, 2.2 Hz, 4H), 6.79 (dd, J = 15.1, 4.7 Hz, 1H), 6.09 (dt, J = 15.2, 2.0 Hz, 1H), 4.55 − 4.46 (m, 1H), 4.43 (s, 1H), 4.21 − 4.09 (m, 2H), 3.90 − 3.84 (m, 2H), 3.61 − 3.52 (m, 2H), 3.30 − 3.18 (m, 4H), 3.10 − 3.00 (m, 2H), 3.00 − 2.89 (m, 2H), 2.72 − 2.62 (m, 1H), 2.62 − 2.52 (m, 1H), 1.96 − 1.85 (m, 1H), 1.85 − 1.74 (m, 1H). ¹³C NMR (126 MHz, MeOD) δ 173.22, 157.20, 146.25, 140.97, 140.39, 137.06, 133.41, 130.04, 129.19, 128.98, 128.27, 128.22, 128.10,

127.33, 126.73, 125.74, 58.30, 56.95, 54.78, 51.49, 51.41, 49.35, 49.26, 40.71, 37.63, 35.10, 31.56 (Supplementary Fig. 12).

**Synthesis of 212-129.** To a solution of 4-[[Bis[[(1,1-dimethylethoxy) carbonyl]amino]methylene]amino]benzoic acid (120 mg, 0.317 mmol) in DMF (3 ml), EDCI (66 mg, 0.346 mmol), DMAP (18.94 mg, 0.346 mmol) and DIPEA (20 mg, 0.346 mmol) were added and stirred for 10 min under room temperature. **212-127** (174.2 mg, 0.288 mmol) in DMF (1 ml) was added afterwards. The reaction mixture was stirred overnight and purified by HPLC to obtain the desired product **212-129**. Yield = 28.8% ESI-HRMS calcd for $C_{51}H_{63}N_7O_{10}S$ $[(M+H)^+]$: 966.4435 found: 966.4485.

**Synthesis of 212-148.** To a solution of **212-129** (67 mg, 0.069 mmol) in DMF (2 ml), HCl (4 M in 1,4-dioxane, 2 ml) was added. The reaction was stirred for 2 h and purified by HPLC to obtain the desired product **212-148**. ESI-HRMS calcd for $C_{41}H_{48}N_7O_6S$ $[(M+H)^+]$: 766.3387 found: 766.2722. $^1$H NMR (500 MHz, MeOD) δ 8.22 (d, J = 8.2 Hz, 1H), 8.16 – 8.10 (m, 2H), 7.87 – 7.81 (m, 2H), 7.73 – 7.66 (m, 1H), 7.61 (dd, J = 8.4, 7.0 Hz, 2H), 7.40 – 7.35 (m, 2H), 7.24 – 7.19 (m, 6H), 7.14 (td, J = 8.0, 5.3 Hz, 4H), 6.77 (dd, J = 15.1, 4.7 Hz, 1H), 6.07 (dt, J = 15.1, 1.7 Hz, 1H), 4.71 – 4.65 (m, 2H), 4.52 – 4.45 (m, 1H), 4.43 (t, J = 7.9 Hz, 1H), 3.62 – 3.54 (m, 2H), 3.38 – 3.33 (m, 4H), 3.02 (dd, J = 13.4, 8.2 Hz, 1H), 2.94 (dd, J = 13.5, 7.6 Hz, 1H), 2.65 (ddd, J = 14.4, 9.5, 5.4 Hz, 1H), 2.56 (ddd, J = 13.7, 9.3, 7.0 Hz, 1H), 1.95 – 1.85 (m, 1H), 1.84 – 1.73 (m, 1H). $^{13}$C NMR (126 MHz, MeOD) δ 173.22, 157.20, 146.25, 140.97, 140.39, 137.06, 133.41, 130.04, 129.19, 128.98, 128.27, 128.22, 128.10, 127.33, 126.73, 125.74, 58.30, 56.95, 54.78, 51.49, 51.41, 49.35, 49.26, 40.71, 37.63, 35.10, 31.56 (Supplementary Fig. 13).

**Reporting summary**
Further information on research design is available in the Nature Portfolio Reporting Summary linked to this article.

## Data availability
All experimental data are provided in the manuscript. The atomic coordinates and structure factor amplitudes of the TMPRSS2 in complex with nafamostat, camostat, UK-371804 and **212-148**; CTSL in complex with E64d and K777; CTSB in complex with E64d, K777 and **212-148** have been deposited in the Protein Data Bank under accession codes 7XYD, 7Y0E, 7Y0F, 8HD8, 8HET, 8HFV, 8HEI, 8HE9, and 8HEN, respectively. Source data are provided with this paper.

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

## Acknowledgements

We thank the staff members from beamlines BL18U1, BL19U1, BL02U1 and BL10U2 at Shanghai Synchrotron Radiation Facility (SSRF) and beamlines I04 of Diamond Light Source for assistance during data collection, as well as technical and instrumental support from the Discovery Technology Platform of Shanghai Institute for Advanced Immunochemical Studies (SIAIS) and Molecular and Cell Biology Core Facility of the School of Life Science and Technology (SLST), ShanghaiTech University. This research was supported by the National Natural Science Foundation of China (grant No. 92169109 to H.Y., No. 32000111 to Q.Y.); Guangzhou Laboratory (grant No. SRPG22-003 and SRPG22-011 to H.Y., No. SRPG22-003 to L.S.); Science and Technology Commission of Shanghai Municipality (grant No. YDZX20213100001556 and 20XD1422900 to H.Y.); the Ministry of Science and Technology of China (grant No. 2021YFC2302500 to L.S.); Basic and Applied Basic Research

Projects of Guangzhou Basic Research Program (grant No. SL2023A04J00076 to Q. Y.); Shanghai Frontiers Science Center for Biomacromolecules and Precision Medicine of ShanghaiTech University.

## Author contributions

H.Y., Z.R., L.S., B.J., X.C. and L.Z. conceived the project; H.W., X.L., M.S., D.L., Y.D. and C.Z. cloned, expressed, purified and crystallized proteins; H.W., X.L., M.S. and Y.D. collected diffraction data and solved the crystal structures. H.W., X.L., M.S., Y.D. and A.H. performed the enzymatic activity and inhibition assays. Y.W. and J.C. conducted the spike protein cleavage inhibition assays. Q.Y. and J.T. performed the cell-based anti-viral assays; Z.X. and X.Y. contributed to the chemical synthesis of the compounds; H.W., Q.Y., Y.D., Y.Z., L.G., H.C., L.Z., L.S. and H.Y. analyzed and discussed the data; L.S., H.W., Q.Y., X.L., Z.X., M.S., D.L., Y.D. and H.Y.wrote the manuscript with inputs from all the authors.

## Competing interests

The authors declare no competing interests.
