## [Peer Review File · Nature Communications]

Reviewers' Comments:

Reviewer #1:

Remarks to the Author:

The manuscript by Wang et al. describes a drug design study to target two human proteases TMPRSS2 and CTSL/CTSB that are responsible for the SARS-CoV-2 entry to cells. Initially, a high-throughput screening was conducted to discover sufficiently potent inhibitors and their crystal structures were determined at decent resolutions, revealing the inhibitors modes of binding. Both covalent and noncovalent inhibitors were found. In the next step the structure-based drug design approach was used to marry inhibitors of TMPRSS2 and CTSL with the purpose of creating a dual-acting inhibitor. Thus, bispecific compound 212-248 was designed and crystal structures with each of the enzymes provided convincing proof of its mode of binding and inhibition. One reactive moiety of the compound is an ester that binds to TMPRSS2 catalytic Cys, whereas the other is the Michael's reagent that binds to CTSB. X-ray crystallographic structures were done very well, and the electron density maps support the conclusions drawn about the inhibitors' binding. The manuscript is written clearly and should be accepted to Nat Commun.

Reviewer #2:

Remarks to the Author:

This manuscript by Wang and co-workers describes the screening, design, and characterization of dual TMPRSS2 and CTSL/CTSB covalent inhibitors as SARS-CoV-2 entry inhibitors. The authors report the co-crystal structures of both TMPRSS2 and CTSL/CTSB inhibitors, some of which, such as UK-371804, have not been reported before. The idea of dual inhibition of two host proteins mediating viral entry is interesting and of great need to overcome emerging SARS-CoV-2 variants. The manuscript is generally well-written and the figures are clear and aesthetically pleasing. However, the manuscript requires additional evidence to prove that 212-148 acts as a dual inhibitor for TMPRSS2 and CTSL/CTSB in cellular or in vivo models, especially considering its highly metabolically labile ester linker. As such, I would only recommend the acceptance of this manuscript in this high-impact journal with further supporting evidence for cellular or in vivo dual inhibition.

Major comments:

- 1) The authors lack data to prove dual TMPRSS2 and CTSL/CTSB inhibition beyond biochemical assays. Compound 212-148 possesses a labile ester as a linker and warhead for TMPRSS2, and this ester can be rapidly hydrolyzed by esterase within minutes (PMID: 29700501). Esterases are abundant in serum and even in the cell culture medium fetal bovine serum (FBS). As such, it's challenging to determine whether the inhibition of viral replication is due to 212-148, the hydrolyzed molecules, or a combination of both. Furthermore, the relatively weak potency of 212-148 compared to the combination study of nafamostat and K777 also suggests the possibility of 212-148 breakdown.
- 2) Evidence of 212-148 inhibiting spike protein cleavage through TMPRSS2 is required.
- 3) The authors may have already begun this work, but other linkers, such as carbamate, have been validated in FDA-approved drugs like rivastigmine as more stable warheads for serine protease. Since the authors have already established a synthetic method, it should be straightforward to synthesize and test it. Alternatively, a dual inhibitor comprising the noncovalent inhibitor UK-371804 and K777 could also address the potential stability issues of 212-148.

Minor comments:

1. The selectivity index and cellular viability are very confusing. No detailed description is reported in the manuscript. In the experimental section, concentrations are not mentioned. In Extended Data Fig. 6. A-G, it seems the authors tested from 10nM to 1uM, however, in Extended Data Fig. 6.E, the unit suddenly changes to μM . The authors need to provide CC50 data in Fig. 4D and 5F.

2. Discussions are needed to compare with the previously reported co-crystal structures of TMPRSS2 with nafamostat or camostat at the catalytic site (Citation 35).

3. Discussion is needed to compare with other manuscripts that reported dual inhibition, such as citations 32 and 33.

REVIEWER COMMENTS

Reviewer #1 (Remarks to the Author):

“The manuscript by Wang et al. describes a drug design study to target two human proteases TMPRSS2 and CTSL/CTSB that are responsible for the SARS-CoV-2 entry to cells. Initially, a high-throughput screening was conducted to discover sufficiently potent inhibitors and their crystal structures were determined at decent resolutions, revealing the inhibitors modes of binding. Both covalent and noncovalent inhibitors were found. In the next step the structure-based drug design approach was used to marry inhibitors of TMPRSS2 and CTSL with the purpose of creating a dual-acting inhibitor. Thus, bispecific compound 212-248 was designed and crystal structures with each of the enzymes provided convincing proof of its mode of binding and inhibition. One reactive moiety of the compound is an ester that binds to TMPRSS2 catalytic Cys, whereas the other is the Michael’s reagent that binds to CTSB. X-ray crystallographic structures were done very well, and the electron density maps support the conclusions drawn about the inhibitors’ binding. The manuscript is written clearly and should be accepted to Nat Commun.”

Response: Thank you for the above positive comments.

Reviewer #2 (Remarks to the Author):

This manuscript by Wang and co-workers describes the screening, design, and characterization of dual TMPRSS2 and CTSL/CTSB covalent inhibitors as SARS-CoV-2 entry inhibitors. The authors report the co-crystal structures of both TMPRSS2 and CTSL/CTSB inhibitors, some of which, such as UK-371804, have not been reported before. The idea of dual inhibition of two host proteins mediating viral entry is interesting and of great need to overcome emerging SARS-CoV-2 variants. The manuscript is generally well-written and the figures are clear and aesthetically pleasing.

Response: We thank the reviewer for recognizing the significance of our research.

However, the manuscript requires additional evidence to prove that 212-148 acts as a dual inhibitor for TMPRSS2 and CTSL/CTSB in cellular or in vivo models, especially considering its highly metabolically labile ester linker. As such, I would only recommend the acceptance of this manuscript in this high-impact journal with further supporting evidence for cellular or in vivo dual inhibition.

Major comments:

1) The authors lack data to prove dual TMPRSS2 and CTSL/CTSB inhibition beyond biochemical assays. Compound 212-148 possesses a labile as a linker and warhead for TMPRSS2, and this ester can be rapidly hydrolyzed by esterase within minutes (PMID: 29700501). Esterases are abundant in serum and even in the cell culture medium fetal bovine serum (FBS). As such, it's challenging to determine whether the inhibition of viral replication is due to 212-148, the hydrolyzed molecules, or a combination of both. Furthermore, the relatively weak potency of 212-148 compared to the combination study of nafamostat and K777 also suggests the possibility of 212-148 breakdown.

Response: We thank the reviewer for the constructive suggestions. Actually, an ethyl linker was employed to connect the pharmacophoric groups of nafamostat and K777 in the design of our bispecific inhibitor 212-148 (Fig. R1a). The ester group of 212-148 as noted in the comment is inherited from the guanidinobenzoate moiety of nafamostat, an FDA-approved drug repurposed as an anti-SARS-CoV-2 agent in phase II clinical trials. The ester group plays a vital role in the covalent reaction

with TMPRSS2's catalytic serine, which was validated in our structural studies. Additionally, this covalent binding motif is also present in camostat. We apologize for this confusion regarding the fusion linker. In the revision, we have changed the corresponding statement to avoid ambiguity (Lines 276-280).

Nonetheless, we fully understand your concern about the potential hydrolysis of 212-148. According to your suggestions, we performed a time-dependent degradation study using LC-MS to examine the stability of 212-148 under the conditions used in the cell assays. Notably, the results showed that more than 50% of the 212-148 remains after two days in the medium that contains FBS (Fig. R1b). In addition, we also show that 212-148 is more stable than nafamostat under the same conditions. Therefore, 212-148 is a proof-of-principle compound that blocks two independent SARS-CoV-2 entry pathways simultaneously. Certainly, we do realize that it shows some defects and will require further optimizations for clinical applications, but such work would be more appropriate as a follow-up manuscript. Overall, while 212-148 is still in its early stages, we think it does represent a viable new structure-based drug discovery strategy for anti-SARS-CoV-2 therapies.

The specific revisions of our manuscript are listed below:

Original: "Inspired by the crystal structures of K777 and nafamostat complexed with their respective proteases, we devised a bispecific compound, named 212-148, by linking the covalent moieties of K777 and nafamostat using an ethyl linker (Fig. 5a)."

Revised: "In light of the crystal structures of K777 and nafamostat complexed with their respective proteases, we devised a bispecific compound, named 212-148. An ethyl linker was utilized to construct this bispecific compound by connecting the pharmacophoric groups of K777 and nafamostat (Fig. 5a)."

Figure R1. Time-dependent degradation analysis of 212-148 and nafamostat by LC-MS. (a) Structure of 212-148 and its specific pharmacophoric groups targeting TMPRSS2 and CTSB/CTSL,

respectively. (b) 125 μM 212-148 or nafamostat was incubated in different conditions, including enzymatic buffer, MEM, MEM w/ 2% (v/v) FBS, MEM w/ 10% (v/v) FBS, DMEM, DMEM w/ 2% (v/v) FBS and DMEM w/ 10% (v/v) FBS, respectively. The LC-MS technique was used to monitor the degradation profiles of the compounds. Specifically, the measurements were conducted daily. The retention content was normalized by comparing it to the original LC-MS measurement value of 212-148 or nafamostat without incubation. The experiments were independently replicated three times. The normalized content is represented as the mean value and shown in the form of histograms. The error bar reflects the standard deviation. Enzymatic buffer: 20 mM Tris pH 7.4, 150 mM NaCl; DMEM: Dulbecco's Modified Eagle Medium used in the Huh-7 cells culture (Hyclone, SH30022.01); MEM: Minimum Essential Medium used in the Calu-3 cells culture (Procell, PM150467); FBS: fetal bovine serum (Gibco, 10099141C).

2) Evidence of 212-148 inhibiting spike protein cleavage through TMPRSS2 is required.

Response: We performed a series of experiments to verify the ability of 212-148 to inhibit spike protein cleavage through TMPRSS2. In order to create the appropriate cleavage assays, we chose the spike protein of the SARS-CoV-2 Delta variant, which predominantly relies on the TMPRSS2-mediated entry pathway.

Firstly, we engineered a stable form of the spike protein by introducing a mutation ($_{681}\text{RRAR}_{685} \rightarrow$ $_{681}\text{GSAS}_{685}$) at the S1/S2 site. This mutation improves the stability of spike protein and ensures a sufficient yield for analysis. Our results demonstrate the successful cleavage of the spike protein at the S2' site by TMPRSS2, yielding two major fragments (Fig. R2a). In contrast, when TMPRSS2 was pre-treated with 212-148, a remarkable inhibitory effect was observed. The spike protein remained uncleaved even after one hour, which is consistent with the findings in the nafamostat-treated positive control group (Fig. R2b).

In parallel, we obtained the spike protein with the original S1/S2 site for cleavage assays. Notably, most spike protein was cleaved into S1 and S2 during cell expression with a small portion of uncleaved spike. Similarly, the cleavage of the original spike by TMPRSS2 was also inhibited by 212-148 or nafamostat treatments, validating their effectiveness in hindering spike protein cleavage (Fig. R2c-d).

In light of these findings, we have validated 212-148's ability to inhibit spike protein cleavage through TMPRSS2. We have incorporated these results into the revised manuscript (Lines 283-285 and Supplementary Fig. 8).

The specific revisions of our manuscript are listed below:

Original: "As expected, 212-148 not only showed potent inhibition for CTSL/CTSB with respective IC_{50} values of 2.13 and 64.07 nM but also suppressed TMPRSS2 enzymatic activity with an IC_{50} value of 1.38 μM (Fig. 5b-c). The designed compound 212-148 conferred the dual-inhibition effects on the biochemical assays, indicating that our drug discovery strategy of targeting two entry pathways is valid."

Revised: "As expected, 212-148 not only showed potent inhibition for CTSL/CTSB with respective IC_{50} values of 2.13 and 64.07 nM but also suppressed TMPRSS2 enzymatic activity with an IC_{50} value of 1.38 μM (Fig. 5b-c). Further, 212-148 demonstrated effective inhibition of SARS-CoV-2 Delta variant spike protein cleavage in vitro, albeit with a decreased anti-TMPRSS2 inhibitory potency (Supplementary Fig. 8). Therefore, the designed compound 212-148 conferred the dual-inhibition effects on the biochemical assays, indicating that our drug discovery strategy of targeting two entry pathways is valid."

Figure R2. Inhibition of TMPRSS2 cleavage of spike protein by 212-148. Time-course cleavage inhibition assay of the mutated spike protein lacking the S1/S2 site (a-b) and original spike protein (c-d) of the SARS-CoV-2 Delta variant. Wherein nafamostat served as a positive control, and SARS-CoV-2 M^{pro} served as a negative control in these assays. Spike protein is indicated by the black arrows, and cleavage products are indicated by the orange and green arrows.

3) The authors may have already begun this work, but other linkers, such as carbamate, have been validated in FDA-approved drugs like rivastigmine as more stable warheads for serine protease. Since the authors have already established a synthetic method, it should be straightforward to synthesize and test it. Alternatively, a dual inhibitor comprising the noncovalent inhibitor UK-371804 and K777 could also address the potential stability issues of 212-148.

Response: We concur that introducing a carbamate group as a warhead could offer enhanced stability of 212-148. However, based on our current synthetic routes, the guanidinobenzoate moiety of 212-148 is integrated within a one-step reaction, posing the challenge of the replacement of benzoate with carbamate. Such studies will take a substantial amount of time and effort, which lies beyond the scope of our current study. As we noted in the response above, we believe the chemical modifications story will require a complete and comprehensive array of work better suited for a follow-up manuscript. Given the biochemical, structural and antiviral achievements regarding the SARS-CoV-2 entry inhibitors discovery, as well as the effort required for the synthesis and evaluation of this new dual inhibitor, we hope that the reviewer is understanding.

Minor comments:

1. *The selectivity index and cellular viability are very confusing. No detailed description is reported in the manuscript. In the experimental section, concentrations are not mentioned. In Extended Data Fig. 6. A-G, it seems the authors tested from 10nM to 1uM, however, in Extended Data Fig. 6.E, the unit suddenly changes to μM. The authors need to provide CC50 data in Fig. 4D and 5F.*

Response: We are sorry for the confusion. The selectivity index (SI) is determined by the ratio of the CC₅₀ value to the EC₅₀ value, which measures the window between cytotoxicity and antiviral activity. Theoretically, the higher the SI ratio, the more effective and safer a drug will be. A detailed description of the selectivity index is included in the supplementary section in the revised manuscript (Lines 244-249). Additionally, we revised our antiviral data after careful data reprocessing. In the original version of the manuscript, we selected one replicate to serve as the representative for evaluation. In the current version, we have taken all replicates into account when calculating the EC₅₀ values. It is worth mentioning that the changes in EC₅₀ and SI values after reprocessing are minor or better (for K777) and do not affect the validity of our conclusions. Furthermore, according to your suggestions, the concentration unit in Extended Data Fig. 6A-G has been corrected to μM, consistent with Extended Data Fig. 6H (now renamed as Supplementary Fig. 6a-h in the revised manuscript). CC₅₀ data has now been provided in Fig. 4D and 5F (now renamed as Fig. 4d and 5f in the revised manuscript).

The specific revisions of our manuscript are listed below:

Original: “K777 and E64d, as endocytosis mediated entrance pathway inhibitors, could effectively reduce the infectivity of the SARS-CoV-2 Omicron BA.2 variant in Calu-3 cells with an EC₅₀ value of 77.7 nM (SI = 1,411) and 227.2 nM (SI > 4,400).”

Revised: “K777 and E64d, as endocytosis mediated entrance pathway inhibitors, could effectively reduce the infectivity of the SARS-CoV-2 Omicron BA.2 variant in Calu-3 cells with an EC₅₀ value of 14.0 nM (Selectivity index (SI) = 7,829, SI is calculated as the ratio of CC₅₀ value against EC₅₀ value, a common measurement for comparing cytotoxicity and antiviral potency of compounds) and 239.8 nM (SI > 4,000).”

2. *Discussions are needed to compare with the previously reported co-crystal structures of TMPRSS2 with nafamostat or camostat at the catalytic site (Citation 35).*

Response: The structures of TMPRSS2 individually in complex with nafamostat and camostat in our study, and a previously reported structure of TMPRSS2 in complex with nafamostat, demonstrate a high degree of similarity, with the r.m.s.d values ranging from 0.296 to 0.445 Å for all Cα atoms in their SP domains (residues 256-492). This makes sense because the same remnant moiety of guanidinobenzoyl group occupied at the catalytic site of TMPRSS2 after the reaction, although the original structures of nafamostat and camostat are different. We have included additional discussion in the revised manuscript (Lines 332-340).

The specific revisions of our manuscript are listed below:

Original: “The crystal structures of TMPRSS2 in complex with potent inhibitors, as well as the structures of CTSL/CSTB in complex with inhibitors, elucidated the active site pocket of the enzymes and revealed the inhibition mechanism, providing a guide for improving inhibitory activity.”

Revised: “The structures of TMPRSS2 in complex with nafamostat or camostat in our study, and a previously reported structure of TMPRSS2 in complex with nafamostat, demonstrate a high degree of

similarity, with the r.m.s.d values ranging from 0.296 to 0.445 Å for all C α atoms of their SP domains (residues 256-492). The values are low because only the same remnant moiety of guanidinobenzoyl group occupies the active site of TMPRSS2 after the reaction, although the complete structures of nafamostat and camostat are different (Fig. 2d). These TMPRSS2-inhibitor complex structures, as well as the structures of CTSL/CSTB in complex with inhibitors, delineate the active site pocket of the enzymes and reveal the inhibition mechanism, providing a guide for improving inhibition.”

3. Discussion is needed to compare with other manuscripts that reported dual inhibition, such as citations 32 and 33.

Response: We agree and have revised the **discussion** in the revised manuscript (Lines 326-330).

The specific revisions of our manuscript are listed below:

Original: “Strikingly, nafamostat showed a 10-fold increase in effectiveness when combined with K777, proving that dual inhibition of these two pathways simultaneously is a more effective way to block viral infection.”

Revised: “Strikingly, nafamostat showed a 10-fold increase in effectiveness when combined with K777, proving that dual inhibition of these two pathways simultaneously is a more effective way to block viral infection. Moreover, comparable synergistic inhibitory outcomes were observed in the other studies^{32,33} by blocking the proteases involving the viral entry pathways across different types of cells and SARS-CoV-2 variants, which further supports dual inhibition as a potential broad spectrum strategy against SARS-CoV-2 or similar infections.”

Reviewers' Comments:

Reviewer #2:

Remarks to the Author:

The revised manuscript has addressed the major concerns regarding the stability of the dual inhibitor and provided evidence for TMPRSS2 cleavage. I would recommend the acceptance of this manuscript with minor edits.

In Figure S8, please add the concentrations of all the inhibitors used in the assays.

In Figure 2d, the curly arrows should indicate nucleophilic addition, not an SN2 reaction.

For Figure 3b, the curly arrows should represent the Michael addition.

REVIEWER COMMENTS

Reviewer #2 (Remarks to the Author):

The revised manuscript has addressed the major concerns regarding the stability of the dual inhibitor and provided evidence for TMPRSS2 cleavage. I would recommend the acceptance of this manuscript with minor edits.

Response: Thank you for your positive feedback and recommendation for acceptance with minor edits. We have addressed these edits promptly as follows.

In Figure S8, please add the concentrations of all the inhibitors used in the assays.

Response: The specific concentrations of nafamostat and 212-148 used in the assays were supplemented within the caption of Figure S8.

The specific revisions are listed below:

Original: "Time-course cleavage inhibition assay of the mutated spike protein lacking the S1/S2 site (a-b) and original spike protein (c-d) of the SARS-CoV-2 Delta variant. Wherein nafamostat served as a positive control, and SARS-CoV-2 Mpro served as a negative control in these assays. Spike protein is indicated by the black arrows, and cleavage products are indicated by the orange and green arrows."

Revised: "Time-course cleavage inhibition assay of the mutated spike protein lacking the S1/S2 site (a-b) and original spike protein (c-d) of the SARS-CoV-2 Delta variant. Wherein nafamostat served as a positive control, and SARS-CoV-2 Mpro served as a negative control in these assays. Prior to cleavage, TMPRSS2 was preincubated with the 10 times molar 212-148 (15 μ M) or nafamostat (15 μ M) for 1h. Spike protein is indicated by the black arrows, and cleavage products are indicated by the orange and green arrows."

In Figure 2d, the curly arrows should indicate nucleophilic addition, not an SN2 reaction.

Response: Thank you for your constructive suggestions. We have revised the illustration of the reaction mechanism of nafamostat or camostat in Figure 2d to eliminate any ambiguity.

The revised Figure 2d is shown below:

Figure 2d. A likely inhibition mechanism for camostat and nafamostat.

For Figure 3b, the curly arrows should represent the Michael addition.

Response: We appreciate your professional advice. According to your suggestions, a detailed reaction route was provided in Figure 3b to ensure clarity.

The revised Figure 3b is shown below:

Figure 3b. Putative inhibition mechanisms of K777.